# A biogenic secondary organic aerosol source of cirrus ice nucleating particles

Martin J. Wolf [1], Yue Zhang [2,3,4,22], Maria A. Zawadowicz [1,5], Megan Goodell [1], Karl Froyd [6,7], Evelyn Freney [8], Karine Sellegri [8], Michael Rösch [1,9], Tianqu Cui [2,10], Margaux Winter [2,11], Larissa Lacher [12], Duncan Axisa [13], Paul J. DeMott [14], Ezra J. T. Levin [14,15], Ellen Gute [16], Jonathan Abbatt [16], Abigail Koss [17,18], Jesse H. Kroll [17,19], Jason D. Surratt [2,20] & Daniel J. Cziczo [2,19,21 ✉]

Atmospheric ice nucleating particles (INPs) influence global climate by altering cloud formation, lifetime, and precipitation efficiency. The role of secondary organic aerosol (SOA) material as a source of INPs in the ambient atmosphere has not been well defined. Here, we demonstrate the potential for biogenic SOA to activate as depositional INPs in the upper troposphere by combining field measurements with laboratory experiments. Ambient INPs were measured in a remote mountaintop location at −46 °C and an ice supersaturation of 30% with concentrations ranging from 0.1 to 70 L$^{-1}$. Concentrations of depositional INPs were positively correlated with the mass fractions and loadings of isoprene-derived secondary organic aerosols. Compositional analysis of ice residuals showed that ambient particles with isoprene-derived SOA material can act as depositional ice nuclei. Laboratory experiments further demonstrated the ability of isoprene-derived SOA to nucleate ice under a range of atmospheric conditions. We further show that ambient concentrations of isoprene-derived SOA can be competitive with other INP sources. This demonstrates that isoprene and potentially other biogenically-derived SOA materials could influence cirrus formation and properties.

[1] Department of Earth, Atmospheric, and Planetary Sciences, Massachusetts Institute of Technology, 77 Massachusetts Avenue, Room 54-918, Cambridge, MA 02139, USA. [2] Department of Environmental Sciences and Engineering, University of North Carolina at Chapel Hill, 135 Dauer Drive, 166 Rosenau Hall, Chapel Hill, NC 27599, USA. [3] Aerodyne Research Incorporated, Center for Aerosol and Cloud Chemistry, 45 Manning Road,, Billerica, MA 01821, USA. [4] Department of Chemistry, Boston College, 2609 Beacon Street, Chestnut Hill, MA 02467, USA. [5] Atmospheric Sciences and Global Change Division, Pacific Northwest National Laboratory, 902 Battelle Boulevard, Richland, WA 99354, USA. [6] NOAA Earth System Research Laboratory (ESRL), Chemical Sciences Division, Boulder, CO 80305, USA. [7] Cooperative Institute for Research in Environmental Sciences, University of Colorado, Boulder, CO 80309, USA. [8] Université Clermont Auvergne, CNRS, Laboratoire de Météorologie Physique (LaMP), F-63000 Clermont-Ferrand, France. [9] Institute for Atmospheric and Climate Science, Eidgenössische Technische Hochschule Zurich, Zurich, Switzerland. [10] Paul Scherrer Institute, Laboratory of Atmospheric Chemistry, Villigen, Switzerland. [11] Department of Chemistry and Chemical Biology, Harvard University, Cambridge, MA 02138, USA. [12] Karlsruhe Institute of Technology, Institute of Meteorology and Climate Research (IMK-AAF), Eggenstein-Leopoldshafen, Germany. [13] Droplet Measurement Technologies, Longmont, CO 80503, USA. [14] Department of Atmospheric Science, Colorado State University, Fort Collins, CO 80523, USA. [15] Handix Scientific, Boulder, CO 20854, USA. [16] Department of Chemistry, University of Toronto, Toronto, ON, Canada. [17] Department of Civil and Environmental Engineering, Massachusetts Institute of Technology, 77 Massachusetts Avenue, Room 1-290, Cambridge, MA 02139, USA. [18] Tofwerk USA, 2760 29th St., Boulder, CO 80301, USA. [19] Department of Chemical Engineering, Massachusetts Institute of Technology, 77 Massachusetts Avenue, Room 66-350, Cambridge, MA 02139, USA. [20] Department of Chemistry, University of North Carolina at Chapel Hill, 125 South Road, Chapel Hill, North Carolina 27599, USA. [21] Department of Earth, Atmospheric, and Planetary Sciences, Purdue University, 550 Stadium Mall Drive, West Lafayette, IN 47907, USA. [22] Present address: Department of Atmospheric Sciences, Texas A&M University, 3150 TAMU, College Station, Texas 77843, USA. ✉email: djcziczo@purdue.edu

Aerosol-cloud interactions affect weather and climate by influencing cloud formation, albedo, and precipitation efficiency[1–3]. Atmospheric ice nucleation, for instance, induces over one half of global precipitation and influences the net radiative impact of clouds[4,5]. Whereas liquid water clouds and mixed phase clouds impart a net cooling effect on climate, completely glaciated cirrus clouds can exert a net warming effect by absorbing and reradiating outgoing long wave radiation back towards the Earth's surface[6,7]. However, despite their importance, ice nucleating particles (INPs) are still poorly represented in models and an improved understanding of ice nucleation in the atmosphere is required to reduce biases in weather and climate predictions[8].

Ice nucleation occurs spontaneously in liquid droplets at high ice supersaturations ($\geq 140\%$ $RH_{ice}$) and temperatures below about $-38\,^{\circ}C$[9]. However, atmospheric ice can form at lower relative humidity and/or warmer temperatures through heterogeneous freezing mechanisms. Deposition nucleation is one possible heterogeneous pathway where supersaturated water vapor deposits onto the surfaces of INPs. Nucleation may also initiate as pore-condensation freezing, where liquid water condenses within surface microstructures at subsaturated conditions due to the inverse Kelvin effect[10,11]. A preponderance of evidence from field and modelling studies indicates that deposition nucleation is likely a dominant formation mechanism of cirrus clouds[12–14]. Cirrus have the greatest spatial coverage of any cloud type and uniquely exert a net warming radiative effect, but this warming is sensitive to global INP abundance[7]. The sources and ambient concentrations of INPs in the cirrus regime must therefore be quantified to constrain the impact of aerosol emissions on climate.

Laboratory analyses have shown that a variety of materials can nucleate ice with varied efficiencies at cirrus-relevant conditions, including mineral dust, organics, black carbon, biological particles, and anhydrous salts[15]. Not all particles that nucleate ice in the laboratory are important for atmospheric ice nucleation, since these particles may not be abundant in the ambient atmosphere. Organic aerosols often comprise the highest aerosol number fraction in both the boundary layer and upper troposphere[16,17]. Recent studies have investigated the potential for SOA material and their laboratory proxies to depositionally nucleate ice[18,19]. Results suggest that the phase and chemical composition of organic-rich particles both affect ice nucleation ability. While some crystalline organic compounds promote depositional ice nucleation, other substances require an amorphous, or glassy, phase to trigger ice nucleation[20–22]. A high particle viscosity decreases the diffusion rate of water into an SOA particle, allowing it to remain glassy and potentially promote heterogeneous ice nucleation. Laboratory data have demonstrated that a variety of surrogates for atmospherically abundant SOA are effective depositional INPs[23], yet few studies have characterized the ice nucleation ability of more chemically complex ambient SOA species. Particles containing oxalic acid dihydrate were found to be an effective INP, yet these particles have low atmospheric abundance[24]. Other characterizations of ambient ice residuals suggest particles with SOA material constitutes 14–24% of the number of heterogeneous cirrus INPs[13,25]. Despite this large fraction, there have been few studies that investigate the sources and chemical composition of the ambient SOA activating as INPs[13,25,26]. A comprehensive understanding of cirrus formation is therefore limited by the lack of field data on the concentration and sources of SOA that act as depositional INPs[27]. As a consequence, models face large uncertainty when predicting how changing volatile organic compound emissions will affect cloud properties.

This study presents atmospheric measurements indicating ambient biogenic SOA may be an important source of INPs.

Measurements from continental Europe show that depositional INP concentration is elevated when sampling secondary organic-rich particles. INP abundance is well correlated with both the mass loadings and mass fractions of isoprene-derived SOA, but poorly correlated with the abundance of anthropogenically sourced SOA. Separate analysis of ice residual composition provides direct evidence of isoprene-derived INPs. Laboratory experiments further quantify the ice nucleation ability of isoprene-derived SOA in the cirrus cloud regimes. Finally, we estimate ambient concentrations of isoprene-derived INPs in the cirrus cloud temperature and supersaturation regime. We show particles with isoprene-derived SOA material are likely abundant enough to affect cirrus formation in the free troposphere. This demonstrates that changing isoprene and other biogenic volatile organic compound emissions may impact the global climate system.

## Results and discussion

**Ambient aerosol and INP measurements**. Here we show a time series of ambient depositional INP measurements taken at the Puy de Dôme observatory in France. Details of measurements and experimental protocol are included in Materials and Methods, with additional description in Supplementary Methods. INP concentration ($T = -46\,^{\circ}C \pm 0.5$; $RH_{ice} = 130\% \pm 3$) spans three orders of magnitude, from $\sim 0.1$ to $70\ L^{-1}$ (Fig. 1a). There is no apparent relationship between INP concentration and ambient meteorology (Supplementary Fig. 1).

Average aerosol particle size distributions during INP sampling are shown in Fig. 1b. Sampling periods with abundant Aitken and small accumulation mode particles (diameters < 150 nm) correlate ($R^2 = 0.88$; $p$ value $1.93 \times 10^{-5}$) with periods with high average INP concentrations (Supplementary Fig. 3; Supplementary Table 1). Conversely, average INP concentration is uncorrelated ($R^2 = 0.06$; $p$ value 0.46) with ambient concentrations of larger mode particles (diameters > 150 nm). Sampling periods with abundant small (diameters < 150 nm) particles correspond to more abundant organic aerosol mass (Fig. 1a upper panel). In particular, elevated INP concentrations on October 5, 6, 7, 10, 11th, 14, and 15th coincided with periods when organics constituted greater than 60% of total non-refractory aerosol mass. While these links alone do not demonstrate the SOA are activating as INPs, it is consistent with a small and organic particle source of INPs.

We alternatively sampled INPs directly from ambient air and from an aerosol concentrator (Materials and Methods). Figure 1c illustrates the average INP diameter for each day derived from the concentrator enrichment factor (Supplementary Fig. 2, Supplementary Methods). Uncertainty derives from variability in the size dependence of the concentration enhancement factor between calibrations. The average INP diameter on days with elevated organic mass fractions was smaller ($75 \pm 24$ nm) than on days with relatively smaller organic mass fractions ($230 \pm 42$ nm). The small apparent size of INPs on many days may be due to an SOA source. Further, average INP diameter was statistically anticorrelated ($R^2 = 0.68$; $p$ value 0.002) with ambient non-refractory organic mass fraction (Supplementary Fig. 3; Supplementary Table 2). This indicate INP diameter was smaller on days with abundant SOA.

**SOA composition and INP concentration**. Aerosol filter samples were taken concurrently during INP measurements between 7 October 7 and 15 October. These samples were analyzed for bulk particle composition (see Materials and Methods). The average aerosol mass loadings during these sampling measurements were quantified for eight SOA species. Components of

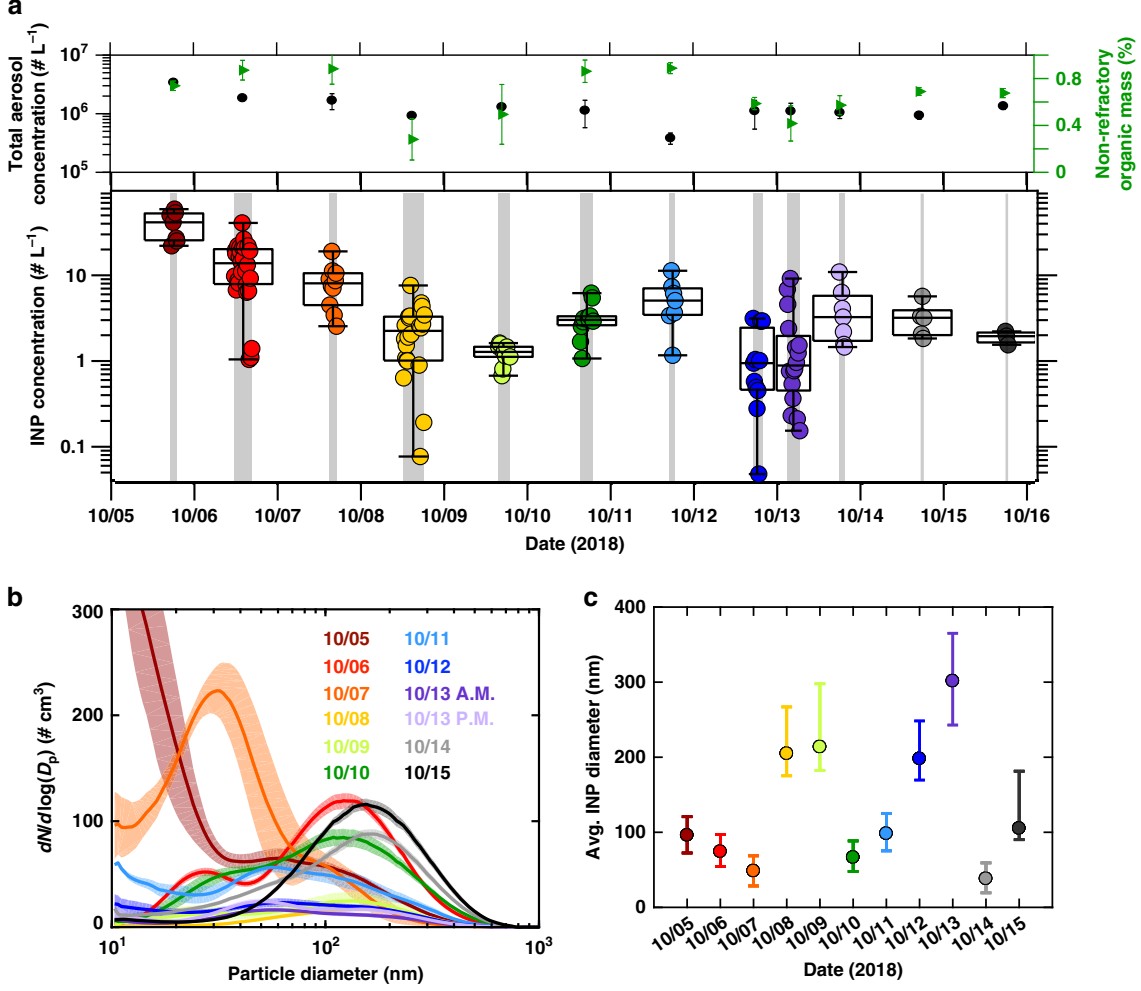

**Fig. 1 Ambient aerosol particle abundance. a** A time series of depositional ice nucleating particle (INP) concentration (−46 °C; $S_{ice} = 1.3$) and averages of total aerosol concentration and aerosol non-refractory organic mass fraction during INP sampling. Each datapoint represents the average ambient INP concentration over 10 min of measurement. Overlain box plots indicate the median, 25th and 75th percentiles, and upper and lower bounds of INP concentrations for each day. Shaded regions correspond to periods of INP measurements. **b** Average ambient submicron aerosol size distributions during INP measurements each day. Shading indicates a standard deviation of concentration variability. **c** Daily average depositional INP derived diameters. The illustrated range is a standard deviation of uncertainty derived from variability in the size dependence of the aerosol concentrator enrichment factor[66].

isoprene-derived SOA[28–31] included first-generation multiphase chemical products of isoprene epoxidiols (IEPOX), specifically 2-methyltetrols (2-MTs, $C_5H_{12}O_4$), 2-methyltetrol organosulfates (2-MT OSs, $C_5H_{12}O_7S$)[32], as well as heterogeneous hydroxyl (OH) radical oxidation products of particulate 2-MT OSs[33], including 2-methylglyceric acid organosulfate (2-MG OS, $C_4H_8O_7S$), and two organosulfates (OSs) with molecular weights of 212 and 214 (OS 212, $C_5H_8O_7S$; OS 214, $C_5H_{10}O_7S$); α-pinene SOA[34] included terebic acid ($C_7H_{10}O_4$); and anthropogenic SOA[35,36] included 4-nitrophenol (4-NP, $C_6H_5NO_3$) and 4-nitro-1-naphthol (4-NN; $C_{10}H_7NO_3$). Measurements of gas-phase chemistry on-site, including ozone, sulfur dioxide, and nitrogen oxide concentrations, show little correlation with INP concentration averaged over daily measurement periods. (Supplementary Table 3; Supplementary Fig. 4). Time of Flight Aerosol Chemical Speciation Monitor (ToF-ACSM) measurements (Material and Methods) also do not indicate correlation with bulk inorganic or total organic mass concentrations (Supplementary Fig. 5) even though INP concentrations were higher on days with more organic aerosol (Fig. 1a).

The average INP concentration during aerosol filter sampling correlates ($R^2 \geq 0.88$) with mass loadings of isoprene-derived

SOA (Fig. 2a). Trendlines indicate ordinary least squares linear regressions. These are calculated using the filter-derived SOA mass loadings and the average INP concentration during filter collection. Regression coefficients and *p* values for these least squares regressions are found in Table 1. Also illustrated via shading around trendlines are one standard error of uncertainty in the slope and intercept of the linear model. High mass loadings of isoprene-derived SOA coincide with abundant Aitken and small accumulation mode particles (Fig. 1b), suggesting these smaller size particles were from SOA sources. The correlation between INP concentration and SOA mass loadings are less strong for α-pinene derived SOA ($R^2 = 0.64$). Anthropogenic-sourced SOA appears uncorrelated with INP concentration ($R^2 \leq 0.12$). Similar correlations exist when the SOA mass loading is normalized to total aerosol burden (Fig. 2b). Total aerosol burden is derived from ambient aerosol size distribution data (Fig. 1b) assuming an average particle density of 1.25 g cm$^{-3}$. Average INP concentrations and isoprene-derived SOA mass loadings were notably higher on 7 October than other days. Since outliers can bias regressions and skew interpretations of atmospheric measurements[37], we also tabulated regression coefficients for linear models derived with all datapoints except the high outlier

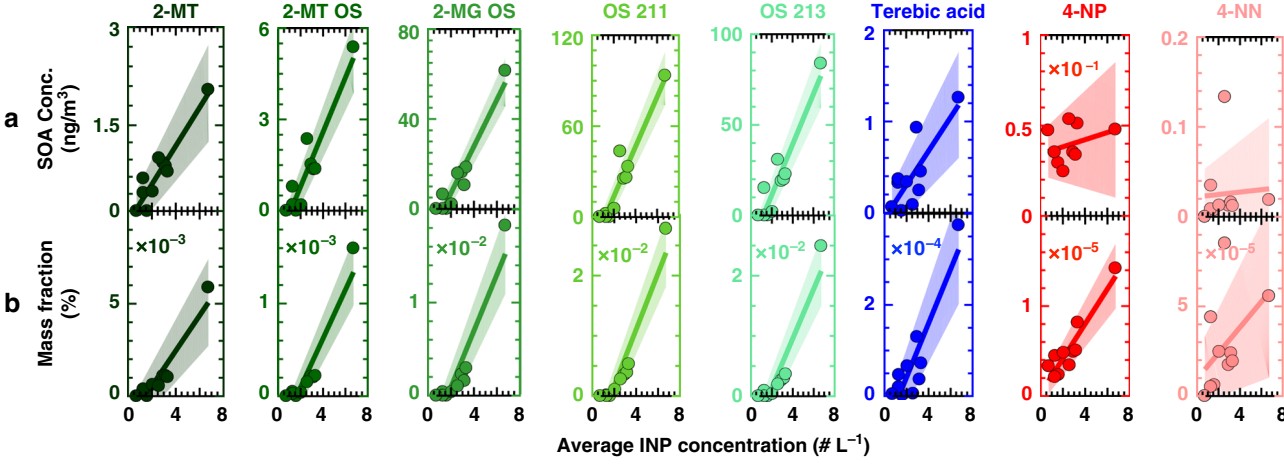

**Fig. 2 Correlations with ice nucleating particle concentration. a** Correlations with secondary organic aerosol (SOA) concentrations are shown for eight species. SOA derivatives of isoprene, α-pinene, and anthropogenic volatile organic compounds are illustrated in green, blue, and red, respectively. Reported ice nucleating particle (INP) concentrations are the average values during filter measurements. Shading represents the standard error of the linear regression. **b** The aerosol mass fractions for the 8 SOA components are plotted against average INP concentration.

**Table 1 Regression coefficients for ice nucleating particles and secondary organic aerosol (SOA) concentrations.**

| SOA component | SOA concentration | | Mass fraction | |
|---|---|---|---|---|
| | $R^2$ ($R^2$ outlier omitted) | $p$ value ($p$ value outlier omitted) | $R^2$ ($R^2$ outlier omitted) | $p$ value ($p$ value outlier omitted) |
| Isoprene | | | | |
| 2-MT | 0.88 (0.62) | $5.4 \times 10^{-5}$ ($1.2 \times 10^{-2}$) | 0.89 (0.83) | $4.5 \times 10^{-5}$ ($5.9 \times 10^{-4}$) |
| 2-MT OS | 0.88 (0.56) | $7.0 \times 10^{-5}$ ($2.1 \times 10^{-2}$) | 0.87 (0.88) | $9.6 \times 10^{-5}$ ($2.1 \times 10^{-4}$) |
| 2-MG OS | 0.92 (0.73) | $1.1 \times 10^{-5}$ ($3.6 \times 10^{-3}$) | 0.86 (0.79) | $1.3 \times 10^{-4}$ ($1.3 \times 10^{-3}$) |
| OS 211 | 0.91 (0.72) | $1.5 \times 10^{-5}$ ($4.1 \times 10^{-3}$) | 0.89 (0.91) | $3.7 \times 10^{-5}$ ($7.7 \times 10^{-5}$) |
| OS 213 | 0.88 (0.56) | $5.9 \times 10^{-5}$ ($2.0 \times 10^{-2}$) | 0.85 (0.86) | $1.2 \times 10^{-4}$ ($4.3 \times 10^{-4}$) |
| α-pinene | | | | |
| Terebic acid | 0.64 (0.20) | $5.6 \times 10^{-3}$ ($2.2 \times 10^{-1}$) | 0.81 (0.30) | $3.9 \times 10^{-4}$ ($1.3 \times 10^{-1}$) |
| Anthropogenic | | | | |
| 4-NP | 0.12 (0.03) | $3.3 \times 10^{-1}$ ($6.5 \times 10^{-1}$) | 0.85 (0.49) | $1.3 \times 10^{-4}$ ($3.5 \times 10^{-2}$) |
| 4-NN | 0.00 (0.05) | $8.7 \times 10^{-1}$ ($5.8 \times 10^{-1}$) | 0.21 (0.09) | $1.8 \times 10^{-1}$ ($4.2 \times 10^{-1}$) |

(Table 1, in parentheses). Although this results in somewhat lower regression coefficients, $p$ values (<0.05) for $R^2$ between INP concentration and isoprene-derived SOA mass loading and fraction indicate statistically significant correlations (Table 1).

Overall, the strongest relationships occur between INP concentration and isoprene-derived SOA mass fraction, yet INP concentration is also moderately correlated with α-pinene SOA mass fraction and concentration (Table 1). Previous studies have indicated α-pinene SOA material can promote heterogeneous ice nucleation at cirrus conditions similar to those considered here[19,20]. However, the correlation coefficient and $p$ value suggest a weaker and statistically insignificant correlation after excluding the outlying datapoint from 7 October. This demonstrates that α-pinene SOA may not have been an important INP source during our measurements. Similarly, the correlation coefficient between 4-NP and INP concentration is only 0.49 after excluding the outlying datapoint (Table 1). The observed correlation could also be due to co-occurrence between different SOA types. For instance, the mass fraction of terebic acid and 4-NP correlate more strongly ($R^2 \geq 0.90$) with the mass fractions of the isoprene-SOA markers than with INP concentration.

One feature of these relationships is the persistent presence of INPs even at low SOA concentrations. This observation is illustrated by an INP concentration of $0.5–1.5\ L^{-1}$ even when SOA was not observed (Fig. 1b) and SOA mass loading was below the detection limit. This is consistent with sampling diverse sources of INPs. INPs aside from SOA, such as mineral dust or other primary aerosols[15], likely constituted the baseline INP concentration during these periods. INP concentrations were enhanced as biogenic SOA became more abundant.

SOA are commonly understood to require an ultraviscous, or glassy, phase state to potentially activate as depositional INPs[21]. Particle viscosity and phase state is determined by composition, degree of oxidation, relative humidity, and temperature[38–40]. Previous studies have shown that the predominant components of isoprene-SOA, specifically IEPOX -derived SOA, are sufficiently viscous to permit heterogeneous ice nucleation at the cirrus conditions investigated here (T = −46 °C; $S_{ice} = 1.3$)[41,42]. We estimate aged 2-MT OSs—major components of IEPOX SOA —to have a glassy transition temperature of $237^{+10}_{-16}$ K and a viscosity of value of $10^{12^{+0}_{-2}}$ Pa s at these conditions (Supplementary Methods)[41,43,44]. These results suggest that organic particles containing a significant amount of 2-MT OSs remain in a glassy or semi-solid phase state at conditions requisite for depositional ice nucleation. However, Price et al., also shows that water diffusion could happen much faster than what the Stokes–Einstein equation predicted using the water-soluble SOA material similar to what was used by Renbaum–Wolff et al.[39,45]. Although further studies on the diffusion rate of water within the 2-MT OSs and isoprene-derived SOA are needed,

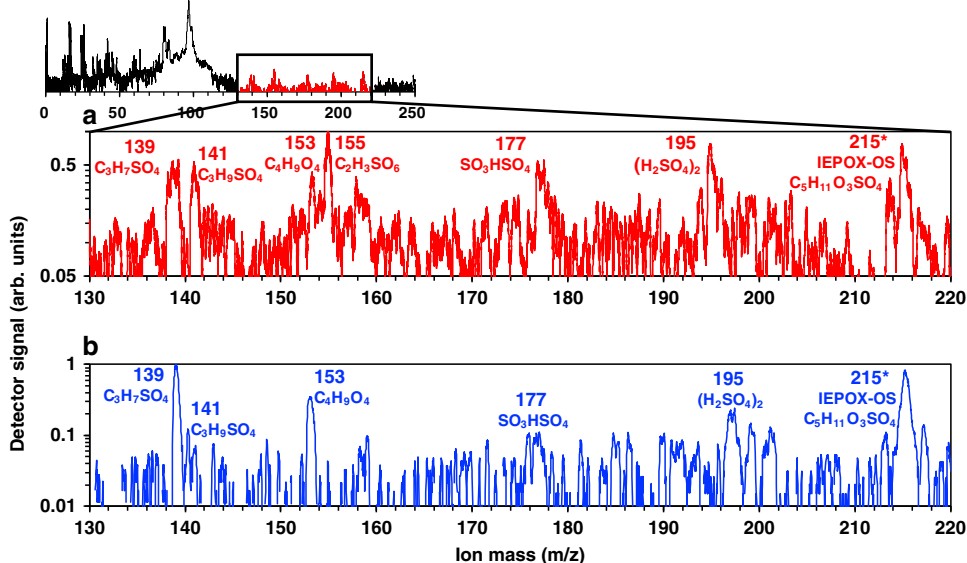

**Fig. 3 Isoprene-derived ice residual analysis.** Spectra show negative ions generated from single particle mass spectrometry (PALMS). Labeled peaks indicate sulfate or organosulfate signatures. Asterisks (*) denote spectral features unique to isoprene-epoxydiol secondary organic aerosol (IEPOX SOA)[47]. **a** Average of ten ice residuals measured under activation conditions in the deposition regime. Measurements were made in Cambridge, MA. **b** Similar markers are observed in a typical spectrum of pure, laboratory-generated IEPOX SOA.

the diffusion rate of water within the 2-MT OSs is estimated to be $\sim 10^{-20^{+2}_{-0}}$ m$^2$ s$^{-1}$ based on the results provided by Price et al.[45]. This yields an average mixing time of $2 \times 10^{8^{+0}_{-2}}$ s (Supplementary Methods). The mixing time of water within the submicron 2-MT OSs particle is much longer than the timeframe for the ice nucleation process[46]. This result suggests that water vapor is unlikely to melt the 2-MT OS particles at cirrus-relevant conditions, and thus supports our evidence that particles containing a significant amount of 2-MT OS or isoprene-derived SOA have the potential to promote heterogeneous ice nucleation.

**Direct atmospheric and laboratory observations.** In separate experiments and at a different sampling location, we measured the composition of depositional ice residuals (IRs) to provide more direct evidence of isoprene-derived INPs. The composition of depositional IR nucleated at −46 °C and an $S_{ice}$ of 1.3 from summertime urban air was measured in real-time using single particle mass spectrometry (Materials and Methods). Ten out of 111 depositional IR collected over four days exhibited features of isoprene-derived SOA, including several OS peaks. One such representative mass spectrum is illustrated in Fig. 3a. Deprotonated ions of these peaks at mass-to-charge (m/z) ratios of 211, 213, and 215 indicates the presence[47] of an isoprene-derived organosulfates (IEPOX-OSs) in the IRs. These are the isoprene-derived species measured at Puy de Dôme (Fig. 2). For comparison, a typical mass spectrum of IEPOX SOA generated in an environmental chamber (Materials and Methods) is shown in Fig. 3b. The overlap of the isoprene-derived OS peak at m/z 215 in both spectra further suggests isoprene-SOA were likely activating as depositional INPs. Notably, OS peaks at m/z 211 and 213 were only observed in the ambient sample, indicating the IEPOX SOA generated in an environmental chamber was not as aged as those in the field. We measured that 9% of depositional IRs contained isoprene-derived SOA material. Moreover, these IRs did not contain signature of refractory material of other known primary INPs, such as mineral dusts, black carbon, and biological particles. The relatively pure SOA spectra demonstrate

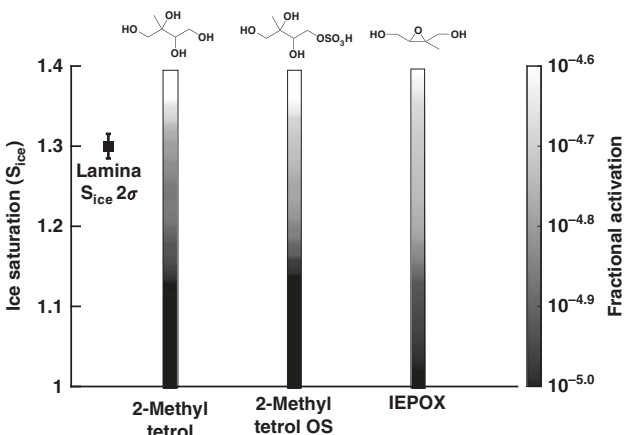

**Fig. 4 Ice nucleation observed in laboratory experiments.** Fractional ice nucleation of particles generated from isoprene-derived components as a function of ice supersaturation in the deposition freezing mode (−46 °C). The data represent the average of four supersaturation scans. For clarity, homogeneous freezing onset above $S_{ice} = 1.4$ is omitted[75]. The left most black datapoint indicates the average variability of $f_{ice}$ at −46 °C and $S_{ice} = 1.3$.

that IEPOX-SOA material from summertime urban air activated as INPs. Although IEPOX-SOA were depleted in IRs relative to bulk ambient aerosol, this is likely due to the low fractional ice activation of organic aerosol (Fig. 4). Selective activation could arise from variation in particle composition, viscosity, and surface properties such as porosity[10,11,21].

We further show that pure components of isoprene-derived SOA compounds nucleate depositionally in laboratory experiments. The fraction of ice-active 2-MTs, 2-MT OSs, and IEPOX particles at −46 °C scales with increasing ice supersaturation (Fig. 4)[29,30,48]. Fractional activation within the deposition freezing regime falls within the range found in other studies on SOA and organic analogs[19,21,49,50]. Many of these studies found that organic particle phase affects ice nucleation ability.

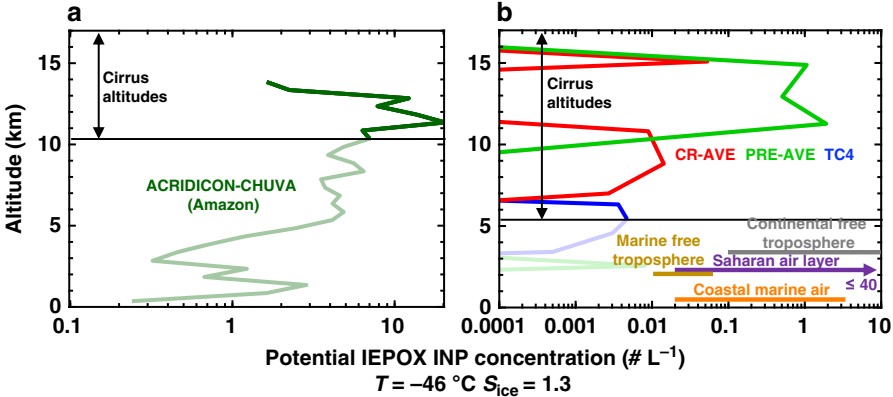

**Fig. 5 Ambient concentrations of ice nucleating particles.** Potential ambient concentrations of isoprene-derived depositional ice nucleating particles (INPs) in **a** a tropical convective outflow system, and **b** typical low and mid-latitude environments. Cirrus altitudes are derived from temperature profiles during the flight campaigns. Data from **a** are derived from the ACRIDICON-CHUVA campaign over the Amazon[51] and data from **b** are derived using single particle mass spectrometry results from the CR-AVE, PreAVE, and TC4 campaigns based in Costa Rica[47] The Pre-AVE campaign was also heavily influenced by Amazonian emissions[52]. For comparison, depositional INP concentrations typical of coastal marine environments[55], the Saharan air layer[54], and the continental[26] and marine[56] free troposphere are also included.

The relatively high glass transition temperatures of 2-MTs, 2-MT OSs, and other IEPOX OS dimers and trimers, as well as the aged IEPOX-derived SOA including 2-MG OS, MW 212 OS, and MW 214 OS, potentially slows water diffusion into the particle. This is one phenomenon that can enhance the ice nucleation activity of SOA material[46]. The observed ice nucleation ability suggests that the physiochemical properties of isoprene-derived OSs may explain their observed ice nucleation abilities. For instance, the formation of IEPOX-OSs is accompanied by the rapid depletion of inorganic sulfates. This reduces hygroscopicity and increases aerosol viscosity[41,44], both of which are factors that could slow the uptake and diffusion of water into isoprene-derived SOA and potentially enhance their depositional ice nucleation properties.

### Atmospheric implications

Our data demonstrate that isoprene-derived SOA compounds, in particular IEPOX SOA, are a potentially important depositional INP source in the continental atmosphere. Aircraft measurements also show that IEPOX-derived SOA account for up to 40% of OA mass in the tropical upper troposphere where cirrus typically form[17,51]. We calculated the potential ambient concentrations of isoprene-sourced depositional INPs in different atmospheric environments (Fig. 5). The potential concentration is higher over the Amazon (Fig. 5a) compared to more diverse tropical locations (Fig. 5b). This is likely due to the elevated isoprene emission rates from forested regions coupled to low $NO_x$ concentrations[52,53]. The average depositional INP concentration measured at the Puy de Dôme during SOA events was $3.8 \pm 1.68 \, L^{-1}$ (1σ; Fig. 1), which is higher than the predicted concentrations for ambient low-latitude environments (Fig. 5b). This implies we sampled multiple sources of depositional INPs and/or sampled concentrations of isoprene-SOA that were higher than typically found in the free troposphere. Nevertheless, the potential INP concentrations from isoprene-derived SOA are comparable to previous measurements of ambient depositional INP concentrations, which can range from 0.1 to $10^3 \, L^{-1}$ [27,54,55]. Figure 5b illustrates typical depositional INP concentrations in marine ($0.2-3.3 \, L^{-1}$)[55] and the mineral-dust rich Saharan air layer (0.2 to $\sim 40 \, L^{-1}$)[54]. Typical depositional INP concentrations in the continental[26] and marine[56] free troposphere range from 0.01 to $10 \, L^{-1}$. This finding indicates isoprene-SOA may also be an important fraction of depositional INPs in the ambient atmosphere, particularly in

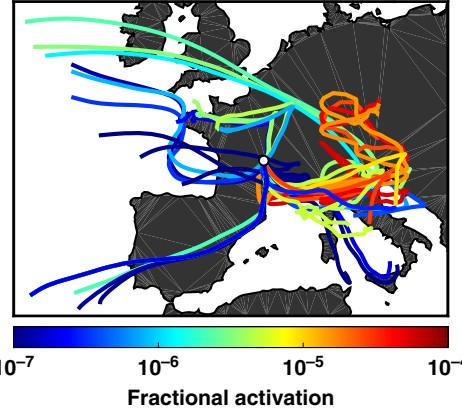

**Fig. 6 Effects of particle source on fractional activation.** Four-day HYSPLIT back trajectories from the altitude of the Puy de Dôme observatory, located at the white dot. Back trajectories were calculated for every hour of sampling. Color indicates the fraction of ambient aerosol activated as depositional ice nucleating particles (INPs). Global Data Assimilation System (GDAS) data at 0.5° × 0.5° spatial resolution were used as meteorological input to calculate the back trajectories.

environments with low abundances of other INPs, such as mineral-dust aerosol[54].

Our finding that isoprene-derived SOA material can promote heterogeneous ice nucleation agrees with previous findings that terrestrial environments are more significant sources of INPs than marine environments[27]. Four-day HYSPLIT air mass back trajectory analyses during the Puy de Dôme measurements show that fractional INP activation is typically elevated when sampling aerosol sourced from forested terrestrial regions over aerosol originating from maritime or agricultural environments (Fig. 6). Only 1 in $10^6-10^7$ aerosol particles from the North Atlantic Ocean or the Mediterranean Sea nucleated ice under depositional freezing conditions. This parallels other findings that air from maritime regions is depleted in INPs[55,56]. While aerosols sourced from Northern France, an area dominated by cropland[57], achieved fractional INP activations up to $7 \times 10^{-6}$, aerosols from heavily forested regions in central and southern Europe are more enriched in INPs (Fig. 6). Above 1 in $5 \times 10^5$ aerosols from these

regions typically activated as INPs. These regions have the highest isoprene emissions rates for continental Europe, which are greater than 0.3 mg m$^{-2}$ h$^{-1}$ and far exceeding emission rates from northern France (<0.05 mg m$^{-2}$ h$^{-1}$) or marine environments, which have low or near-zero isoprene emission rates[58]. The observed enrichment in INPs when sampling air from regions with high isoprene emissions strengthens the link between INP concentration and isoprene-derived SOA abundance demonstrated in Fig. 2. However, we note that the elevated fractional INP activation in air masses passing over Northern Italy may also be due to emissions of mineral dust or anthropogenic INPs, such as black carbon, that have been inferred to activate as INPs at cirrus conditions[15,59].

These results specifically highlight the potential impact of isoprene-derived organosulfate SOA on cirrus cloud formation. Given the rarity of INPs in the ambient atmosphere and the net warming effect of cirrus clouds on global climate, our findings indicate that isoprene-derived INPs may be abundant enough to impact cirrus nucleation. Further work should investigate other SOA-derived sources of ambient INPs. Quantifying the efficacy of these ambient aerosol to nucleate at lower ice supersaturations will help clarify their climatic importance when SOA sources are combined with other effective depositional INP sources. Further studies should also quantify how evolving isoprene and other biogenic VOC emissions may impact INP concentration. Climate, land use, and land cover change have altered historical isoprene emission, resulting in a net isoprene decrease of ~25% since 1900[60,61]. This emissions decline may have simultaneously led to a decrease in INP sourced from isoprene-SOA. Future isoprene-SOA mass loadings may be similarly altered by evolving climate, $CO_2$ concentrations, land use patterns, and sulfate aerosol burdens[60,62,63]. Additional modeling and experimental studies are needed to clarify the implications of these evolving SOA mass loadings for global INP concentrations.

## Materials and methods

**Field site**. Ambient INP concentrations were measured from 5 October through 15th, 2018, at the Puy de Dôme Observatory located in central France (45.772 °N, 2.965 °E). The observatory is 1.47 km above sea level, ~1 km higher than the surrounding topography. It lies 16 km west of the city of Clermont-Ferrand (population 142,000). The experimental setup is illustrated in Supplementary Fig. 6. Aerosols were sampled from a whole air inlet (WAI) atop the laboratory ~15 m above ground level. The WAI has a 50% cutoff diameter of approximately 30 μm (at wind speeds below 10 m s$^{-1}$), allowing both cloud droplets and dry aerosol particles to be sampled. Cloud droplets are subsequently evaporated to cloud residuals prior to being measured. Coarse wire mesh is used to reduce wind speeds near the inlet to ensure efficient sampling at high wind speeds. Diffusion losses in the inlet have been estimated to be <5% for particles larger than 15 nm in diameter and less for smaller particles derived from SOA generation. These design elements allow for sampling aerosol assemblages that are representative of the clear sky aerosol or that expected after the dissipation of a cloud[64]. Aerosols were drawn in through WAIs and sampled with (1) a custom-made scanning mobility particle sizer (SMPS) and a condensation particle counter (CPC; TSI Inc. 3010; lower size limit of 10 nm), which gathered ambient aerosol particle size distribution and concentration data, (2) aerosol particle filters, and (3) the ice nucleation measurement setup.

**INP concentration measurements**. Real-time INP concentrations were measured using the SPectrometer for Ice Nuclei (SPIN, Droplet Measurement Technologies). The theory and operation of SPIN have been described previously[65]. SPIN was operated at an ice saturation ratio ($S_{ice}$) of 1.3 ± 0.05 and a temperature of −46 ± 0.3 °C. Size was used to assign particles as activated INPs. An optical particle counter individually detects particles larger than 500 nm in diameter after the ice nucleation chamber. A 2.5 μm $d_{50}$ size-cut cyclone impactor (BGI Inc., SCC1.062 Triplex) was used upstream of the SPIN inlet to remove lager particles from the sample. Due to nonidealities in the operation of the impactor, particles up to 5.0 μm in diameter entered SPIN. To account for this, only particles larger than 5.0 μm in diameter were counted as activated INPs (Supplementary Fig. 7). We note this is a conservative size limit, and thus our reported INP concentrations may be lower limits.

Particles were sampled through two 30 cm diffusion dryers filled with molecular sieves to prevent frost formation at the SPIN inlet. Nevertheless, frost shedding can occasionally introduce artificial ice counts. To correct for this artefact, 5-min backgrounds were measured every 10 min of sampling by applying a filter upstream of the SPIN inlet (Supplementary Fig. 7). The average apparent INP concentrations for background periods before and after a sampling period were subtracted from the average INP signal measured during sampling. To ensure good signal quality, only data from when the average sampling period INP concentration is twice the average background INP concentration are reported.

Particles were either sampled directly from the WAI or from an aerosol concentrator[66]. The particle enrichment factor is size dependent, ranging from 1 (no concentration enhancement) to 25 for particle diameters between 50 nm and 1 μm, respectively[66]. Because INP size can vary depending on source, an INP-specific enhancement factor ($EF_{INP}$) was determined by alternatively sampling from the concentrator and ambient via the WAI every 60 min. $EF_{INP}$ is calculated as follows:

$$EF_{INP} = \overline{[INP]}_C \times \left( \frac{\overline{[INP]}_{A,t_1} + \overline{[INP]}_{A,t_2}}{2} \right)^{-1}$$

where $\overline{[INP]}_C$ is the time-averaged INP concentration when sampling from the concentrator, $\overline{[INP]}_{A,t_1}$ is the time-averaged INP concentration during the ambient sampling period directly before sampling from the concentrator, and $\overline{[INP]}_{A,t_2}$ is the time-averaged INP concentration during the ambient sampling period directly after sampling from the concentrator. When reporting INP concentration for periods when SPIN sampled from the concentrator, the data are therefore corrected by dividing the measured INP concentrations by that period's $EF_{INP}$. The campaign-averaged $EF_{INP}$ is 5.18 for the conditions considered here. Fractional activation was calculated by dividing the background-subtracted INP concentration by the total ambient particle concentration, as measured by the CPC. INP size was not measured directly. However, the INP-specific enhancement factor is indicative of the average INP diameter (Supplementary Fig. 2; Supplementary Methods).

**Aerosol filter sample analysis**. During INP measurements, bulk aerosols were collected from the WAI for offline chemical analysis. Particles were passed through a 2.5 μm cyclone impactor (URG Corp. Model 2000-30; 16.7 L min$^{-1}$) and collected on 47 mm PTFE membranes (Whatman GE) in inline filter cartridges (Pall, Life Sciences). The airflow rate through each impactor was regulated with a mass flow controller (Alicat MC series) at 16.7 standard liter per minute (slpm). Filter blanks were collected between each sample and uniformly demonstrated that any contamination was below detection limits. The impactors and filters were located beneath the WAI to prevent gravitational settling of particles in tubing. Filters were stored on-site at −20 °C in 10 mL acid precleaned vials and frozen during shipment. The samples were then stored at −80 °C for ~3 months before processing. Briefly, each filter was extracted with 22 mL of high-purity methanol (LC-MS CHRO-MASOLV-grade, Sigma–Aldrich) by sonication for 45 min. Prior to drying, extracts were filtered through 0.2-μm PTFE syringe filters (Pall Life Science, Acrodisc) to remove insoluble species. Nitrogen gas was used to evaporate the solvent from the filtrate and concentrate the soluble components. Particle-phase chemical composition was quantified using hydrophilic interaction liquid chromatography (HILIC)/ESI-HR-quadrupole time-of-flight mass spectrometry (QTOFMS) protocol that can resolve and measure the major IEPOX-derived SOA and other constituents in both laboratory-generated SOA and atmospheric PM with high accuracy. Authentic standards of 2-MTs (~100% purity), 2-MT OSs (86% purity), and 2-MG OS (83.7%) were synthesized at UNC according to previously published procedures[67,68]. Commercially-available standards of terebic acid, 4-nitrophenol, and 4-nitro-1-naphthol were obtained from Sigma-Aldrich (>98% purity). All standards were used to generate calibration curves for the HILIC/ESI-HR-QTOFMS identification and quantification of these organics in aerosol samples collected from the Puy de Dôme Observatory.

**Ice residuals analysis**. The composition of ice residuals in the deposition freezing mode was measured in Cambridge, Massachusetts in August 2017. IEPOX SOA has a unique signature in PALMS mass spectra that enables the unambiguous detection of IEPOX-SOA material in ice residuals. The other SOA sources considered here, such as α-pinene and anthropogenically sourced SOA material, have no identified unique signature. We were therefore unable to detect the presence of these other SOA materials in ice residuals. The methodology has previously been described (e.g., Cziczo et al.[69] and DeMott et al.[26]). Briefly, aerosol particles were passed through a 1.0 μm impactor and drawn into the SPIN to induce ice nucleation at typical cirrus conditions (T = −46 °C; $S_{ice}$ = 1.3). A pumped counterflow virtual impactor (PCVI) was used to isolate ice crystals from unactivated aerosol[70]. The ice crystal size threshold of the PCVI was between 2.5 and 3.0 μm. The ice crystals are sublimated downstream of the PCVI, allowing the Particle Analysis by Laser Mass Spectrometry (PALMS) instrument to analyse ice crystal residual nuclei[71]. Large frost shedding from the SPIN walls can sometimes allow unactivated aerosol to pass through the PCVI, potentially introducing non-IR into PALMS. To correct for these artefacts, an OPC was run in parallel downstream of the PCVI to detect times period of anomalously high particle counts indicative of frost events. PALMS spectra collected during these frost shedding events were discarded.

**Laboratory ice nucleation experiments**. Solutions of authentic 2-MTs and 2-MT OSs were atomized using a TSI Model 3076 Aerosol Generator and then were passed through a diffusion dryer. IEPOX-SOA were generated by atomizing an aqueous solution of ammonium bisulphate and gas-phase IEPOX via a syringe pump and heated inlet to the MIT environmental chamber with 1–3% relative humidity under dark conditions (and thus, no oxidant present). IEPOX (or more specifically trans-\beta-IEPOX, which is the predominant IEPOX isomer in the atmosphere)[72] was synthesized at UNC according to published procedures[73]. The MIT environmental chamber is a 7.5 m$^3$ perfluoroalkoxy bag in a temperature-controlled room that has been previously described and characterized[74]. A polydisperse particle stream was drawn into the SPIN from the environmental chamber. The ice supersaturation was raised from 1.0 to 1.5 isothermally (T = −46 °C) at a rate of 0.02 per minute, then lowering ice supersaturation at an equal rate to ice saturation.

**Calculating potential ambient IEPOX INP concentration**. The potential ambient concentration of IEPOX INP are calculated by multiplying the total ambient IEPOX concentration by a factor that accounts for the low fraction of these particles that activate as INPs. Ambient aerosol are often internally mixed. Our laboratory-generated IEPOX particles were ≥10% IEPOX by mass. On average, 1.93 in 10$^5$ of these particles nucleated ice at typical cirrus conditions (T = −46 °C; S$_{ice}$ = 1.3). This fractional activation (f$_i$) can be applied to ambient number concentrations of aerosol with ≥10% IEPOX by mass to derive potential concentrations of IEPOX INP. The vertical concentration profiles of particles with ≥10% IEPOX by mass are derived from previously reported flight campaigns[47,51] using the method of Froyd et al. 2019. These profiles are then multiplied by f$_i$ to obtain potential INP concentrations. Further details for quantifying IEPOX mass fraction in ambient SOA are detailed in Froyd et al.[47] and Schulz et al.[51].

## Data availability

All data used to generate figures and support the findings of the manuscript's results are available from the corresponding author (djcziczo@purdue.edu) upon request. Measurements at the Puy de Dôme observatory are available at https://icos-atc.lsce.ipsl.fr/dp.

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

## Acknowledgements

We would like to thank the numerous researchers and staff of the Laboratoire de Météorologie Physique at the Université Clermont Auvergne and Daisy Caban for their support during sampling at the Puy de Dôme Observatory. We also thank James Charles Wilson (University of Denver), Luke Ziemba (NASA), and Chuck Brock (NOAA) for their involvement with flight campaign data used to derive ambient INP number concentrations. We also thank Avram Gold (University of North Carolina at Chapel Hill) and Zhenfa Zhang (University of North Carolina at Chapel Hill) for providing authentic standards of 2-MTs, 2-MT OSs, 2-MG OS, and IEPOX. JDS, YZ, and TC thank the U.S. National Science Foundation grant number 1703535 for supporting the synthesis of these standards and for chemical analyses of the ambient PM2.5 samples collected from Puy de Dôme Observatory. YZ was supported by the NSF Postdoctoral Fellowship under AGS Grant no. 1524731 and the National Institutes of Health (NIH) grant no. T32ES007018. JHK acknowledges support from the U.S. National Science Foundation (grant no. AGS-1638672). ARK acknowledges support from the Dreyfus postdoctoral program. The Puy de Dome Research was supported by ACTRIS-2 and the European Commission under the Horizon 2020 Research and Innovation Framework Programme, H2020-INFRAIA-2014-2015, grant agreement number 654109. P.D. and E.L. were partially support by U.S. National Science Foundation grant number 1660486.

## Author contributions

M.J.W. designed and carried out field sampling experiments, with assistance from Y.Z., M.G., E.F., K.S., L.L., D.A., P.J.D., E.J.T.L, E.G., J.A., and D.J.C. Y.Z., M.A.Z., M.R., T.C., M.W., A.K., J.K., J.D.S., and D.J.C. helped with laboratory analyses and sample preparation. K.F. provided analysis of aircraft data. All authors helped in preparation of the manuscript.

## Competing interests

The authors declare no competing interests.
