## [Peer Review File · Nature Communications]

Reviewers' Comments:

Reviewer #1:

Remarks to the Author:

Review of "A Biogenic Secondary Organic Aerosol Source of Ice Nucleating Particles" by M. J. Wolf and colleagues.

The paper addresses the question whether secondary organic aerosol (SOA) can promote ice nucleation. The authors claim that isoprene derived SOA correlates with INP and provide evidence from laboratory experiments that isoprene SOA can form ice. Ice nucleation measurements are technically very challenging and thus still quite rare, which has hindered our understanding INPs and their potential sources. The approach of the study and results are novel and represent an important contribution to the field. The results could potentially be used to improve the representation of aerosol-cloud interactions in weather and climate models. Thus, I find the manuscript in principle suitable for Nature Communications.

The manuscript is well written and easy to follow and the data is presented clearly. I have, however, some important reservations about the quality of the analysis, which must be addressed before the manuscript can be considered for publication.

I'm most worried about the validity of the statistical analyses presented, as the main claim of the article is based on correlations found in Fig 2. It is not explained how the regression lines and coefficients were calculated. It looks like one INP value in Fig 2. is clearly higher than the other values, which probably impacts the results a lot, depending on the method chosen, especially since the data set is very small and there are quite large uncertainty in the INP measurements. The authors should go through and justify the statistical analysis methods of the paper. E.g. Mikkonen et al. (2019) show the impact of measurement uncertainty and outliers on different regression analysis methods using atmospheric data.

What is the second set of regression coefficient values in parenthesis in Table 1? Why is 4-NP mass fraction well correlated with INP, but 4-NP concentration not? My guess is that this shows that you cannot make too bold conclusions based on this data.

The INP number in Fig 1 range from 0-70 #/L, but in Fig 2 and Fig S5 from 0-7#/L. Is only a subset of data presented in Fig2 and S5 or is the data averaged differently? How does it affect your conclusions that the higher INP values are not included in the regression analysis?

The presented data set is very short, only 10 days. From Fig S1, I also understood that INPs were measured only during a short period on each day. It would be good to somehow show the times of the INP measurements also in the main manuscript, from Fig 1 you don't really see it. Is the size distribution in Fig 1b averaged over the period of INP measurements or the whole day?

Figure S1b and S3b. There is rather large variation in INP concentrations during each measurement period, while the meteorological and gas conditions are quite stable (as the time period is rather short), which probably results in these horizontal lines in the scatter plots. It would make sense to plot (and calculate R² values using) the average values of each INP measurement period against the average meteorological and gas concentrations during that time, or possibly show both 10min and sampling period average values. Perhaps using log-scale for INP concentrations would make the 10min plot more readable?

In text you mention, that the INP number is connected to small aerosol number and organic mass fraction, and that INP size with organic mass fraction. Could you show the regression coefficient (maybe even a figure) between these variables (INP number and size with total aerosol number, small aerosol number and organic fraction)?

On page 7, you explain that the trajectories with high INP concentrations are from forested areas with high isoprene emission. From FigS6 it seems that all red colored trajectories pass northern Italy with a lot of anthropogenic pollution as well (although it is hard to tell as the figure is rather small). Therefore, the statement on rows 14-16 might not be justified, as also other terrestrial gas and particle sources from both biogenic and anthropogenic emissions could explain the higher INP concentrations.

Did the measured ice residuals exhibit features of other types of SOA than isoprene-derived SOA, or did you only look for isoprene-SOA? Is there a reason why isoprene derived SOA is better ice nuclei than other SOA? You might want to comment on that in the article.

References:

Mikkonen, S., Pitkänen, M. R. A., Nieminen, T., Lipponen, A., Isokääntä, S., Arola, A., and Lehtinen, K. E. J.: Technical note: Effects of uncertainties and number of data points on line fitting – a case study on new particle formation, *Atmos. Chem. Phys.*, 19, 12531–12543, <https://doi.org/10.5194/acp-19-12531-2019>, 2019.

Reviewer #2:

Remarks to the Author:

General comments:

The manuscript by Wolf et al presents a combined field and laboratory experimental study of ice-nucleating particles (INP) at cirrus conditions. The authors suggest that small organic aerosol particles originating from the atmospheric oxidation of isoprene are responsible for the observed ice-nucleation activity at the Puy de Dome observatory in France, based on aerosol composition and several types of correlations with INP concentrations. They conclude that this may be important for cirrus cloud formation and their climatic effects.

While I agree that there are indeed indications for the fact that isoprene-derived SOA may be a source of INP, the causality and the presented experimental evidence does not fully convince me. I acknowledge the different individual small pieces of evidence that may point in the right direction, but a straightforward and continuous proof is missing in my opinion. I cannot pinpoint to a potential fundamental flaw: I actually fully believe that the presented data have been obtained and analysed according to the highest standards.

So maybe I am too critical, but the manuscript's conclusion appear too speculative to me. Even the authors themselves often use conditional voice, see e.g., the summarizing sentence in the introduction on page 3, lines 44/45: 'This study presents atmospheric measurements indicating ambient biogenic SOA may be an important source of INPs.' Similar types of statements occur on page 4, lines 25/26 and 36/37.

I am not convinced by the explanations and experimental support provided for the suggested climatic importance of isoprene-derived SOA in cirrus clouds. The statement that '... our findings indicate that isoprene-derived INPs play an important role in the radiative budget.' (page 7, lines 19-21) to me is an exaggeration and, thus, highly speculative. Given this fact, I cannot confidently support publication. While the experimental data are indeed useful, the overall picture and interpretation is too weak to allow rating this work as 'representing important advances of significance' and 'providing strong evidence for its conclusions'. For a journal with such a high reputation as Nature Communications I would assume a more convincing line of evidence. I am sorry that I cannot be more positive at this time.

Scientific points:

(1) The strongest data appear to be the correlation between INP concentration and isoprene SOA mass loading presented in Figure 2. However, the pinene-derived SOA correlates strongly with isoprene SOA (actually stronger than with INP), so I was wondering what all these correlations really mean. Is it possible that the correlation with isoprene just indicates that aged INP (of whatever type) are more active than fresh ones, and they just happen to accumulate SOA material during their lifetime, and then preferably isoprene-derived SOA material given that isoprene is one of the most abundant biogenic volatile compounds in the atmosphere?

(2) Page 5, lines 24-31: Apparently, the data also point towards other types of INP being important. Together with the ice residual measurements showing only ~9% of isoprene-related INP, I was wondering how important are the isoprene-derived SOA overall?

(3) Page 5, lines 38/39: 'At these conditions, aged 2-MT OS – a major component of isoprene-derived SOA – has a glassy transition temperature of approximately 3 °C (Ref.40).' As far as I understand the glass transition temperature quoted from Ref 40 refers to the pure component 2-MT OS, i.e. at dry conditions. At the cirrus-type conditions studied here ($S_{ice}=1.3$ corresponding to about RH=84%) the glass transition of 2-MT OS may be significantly lower.

(4) Page 5, line 45 and following: The measurements on ice residuals were actually done in an entirely different (urban) location in a different year than those of the ambient aerosol and INP measurements. Moreover, using data of only 111 collected particles of which 10 exhibited features of isoprene-derived SOA (page 6, lines 1/2) to conclude that this 9% of particles support 'the causality of our correlation between isoprene SOA and depositional INP concentrations measured at the Puy de Dome observatory' appears to me as a bit of a stretch.

Minor and technical remarks:

(5) In general, the manuscript is well written, although I felt the line of thought was somewhat hard to follow: the different pieces of evidence were presented consecutively, but only during the second read, I understood how they were related. Moreover, some of the details in the main text were not required for the general understanding (e.g., lines 29-32 on page 4).

(6) Page 3, line 4: Change 'whereas liquid water and mixed phase clouds' to 'whereas liquid water clouds and mixed-phase clouds'

(7) Page 3, line 37: do the given percentages refer to aerosol number or mass? Please specify.

(8) Page 9, line 39: It appears to me that this reference is not correct, as in that paper the synthesis of 2-MT OS is not described at all, and that of 2-MT only as an intermediate product in another synthesis, i.e. of one of various IEPOX species. Please provide correct reference.

Reviewer #3:

Remarks to the Author:

This is an interesting and exiting paper. Wolf et al. set out field and laboratory evidence that biogenic naturally occurring secondary organic aerosol nucleate ice in the deposition mode whereas anthropogenic organic aerosol does not nucleate ice. This is exciting because it implies that changes in aerosol composition due to climate change, land use change and other human activities may affect the concentration of ice nucleating particles in the upper troposphere which

may then be important for climate. There is evidence in the literature that glassy materials of atmospheric relevance can nucleate ice, but this paper is novel in that it uses data from the real atmosphere. The correlations on their would be interesting, but inconclusive, but together with the size, composition and supporting lab work a convincing case is made. I support the paper's publication in close the current form. I have a few comments which I think will lead to further improvements in the paper:

1. Consider alluding the cloud type this paper is relevant for in the title. When I saw the title I immediately thought about mixed phase clouds (reflecting my bias). But, the idea that biogenic material might impact cirrus clouds is important and distinct, and therefore should be clear in the title. Possibly insert 'cirrus' before 'Ice nucleating particles'.
2. P 5 Ln 33-42 (and elsewhere). Yes, being in a glassy state correlates with a material's ability to nucleate ice. But, it is not the defining physical variable which controls if an aerosol will nucleate ice heterogeneous or if it will swell with water and freeze homogeneously (in fact, the glassy state is arbitrarily defined depending on the physical technique being used to probe a material). The key variable is diffusion of water. The diffusion coefficient determines how rapidly water will diffuse into a droplet, if it is slow relative the change in conditions, then a glassy aerosol will remain largely solid and then has the potential to nucleate ice. Diffusion is related to viscosity in a complex manner and the standard Stokes-Einstein equation breaks down near the glass transition (e.g. see [Price et al., 2016]). So, it is much more accurate to talk about aqueous aerosol with slow water diffusion having the potential to nucleate ice heterogeneous than aerosol with high viscosity or being in a glassy state.
3. There are diffusion and viscosity measurements of the water soluble component of SOA which should be drawn upon for this study [Price et al., 2015; Renbaum-Wolff et al., 2013].
4. P5. Ln 25. It is not clear what flattening out means when viewing these plots. I think it is simply that the Number of ice crystals does not tend to zero as SOA tends to zero.
5. P7. Ln 8. Ref 30 is a paper about mixed phase conditions and the comparison with the cirrus conditions made here should not be done. INP in the cirrus and mixed phase regimes will only sometimes be correlated.
6. Could the authors suggest what might happen to alpha-pinene SOA in the future and what has happened to the balance between biogenic and anthropogenic SOA in the past? What are the implications for ice nucleation?
7. Fig 1b. Is there a reason for plotting the size distribution in N rather than the more normal $dN/d\log D_p$?
8. Consider bringing S4 and S6 into the main paper. I think the formatting would allow it and I think these plots are sufficiently important to the paper that they should be there.

Refs.

Price, H. C., J. Mattsson, and B. J. Murray (2016), Sucrose diffusion in aqueous solution, *Phys. Chem. Chem. Phys.*, 18(28), 19207-19216, doi:10.1039/c6cp03238a.

Price, H. C., et al. (2015), Water diffusion in atmospherically relevant [small alpha]-pinene secondary organic material, *Chemical Science*, 6(8), 4876-4883, doi:10.1039/c5sc00685f.

Renbaum-Wolff, L., J. W. Grayson, A. P. Bateman, M. Kuwata, M. Sellier, B. J. Murray, J. E. Shilling, S. T. Martin, and A. K. Bertram (2013), Viscosity of α -pinene secondary organic material and implications for particle growth and reactivity, *P. Natl. Acad. Sci. USA*, doi:10.1073/pnas.1219548110.

Dear Editor,

We would like to thank the reviewers for their careful reading of this manuscript and their suggestions to improve it. In response to reviewer comments, we have substantially revised the manuscript. We believe the new manuscript addresses reviewer concerns and as a result is scientifically improved.

Our point-by-point response to reviewer comments is included below. For clarity, reviewer comments are in green text and our responses are in black. Our line numbers reference the updated document, not the original.

Reviewer 1

1. The paper addresses the question whether secondary organic aerosol (SOA) can promote ice nucleation. The authors claim that isoprene derived SOA correlates with INP and provide evidence from laboratory experiments that isoprene SOA can form ice. Ice nucleation measurements are technically very challenging and thus still quite rare, which has hindered our understanding INPs and their potential sources. The approach of the study and results are novel and represent an important contribution to the field. The results could potentially be used to improve the representation of aerosol-cloud interactions in weather and climate models. Thus, I find the manuscript in principle suitable for Nature Communications.

The manuscript is well written and easy to follow and the data is presented clearly. I have, however, some important reservations about the quality of the analysis, which must be addressed before the manuscript can be considered for publication.

We thank the reviewer for their thorough consideration of our manuscript and for reiterating the potential significance of our findings. In our responses below, we address the reviewer's reservations about our study and highlight changes made to improve the clarity and statistical rigor of our data analyses.

2. I'm most worried about the validity of the statistical analyses presented, as the main claim of the article is based on correlations found in Fig 2. It is not explained how the regression lines and coefficients were calculated. It looks like one INP value in Fig 2. is clearly higher than the other values, which probably impacts the results a lot, depending on the method chosen, especially since the data set is very small and there are quite large uncertainty in the INP measurements. The authors should go through and justify the statistical analysis methods of the paper. E.g. Mikkonen et al. (2019) show the impact of measurement uncertainty and outliers on different regression analysis methods using atmospheric data.

We thank the reviewer for pointing out the ambiguity of the statistical analyses of Figure 2. The reviewer correctly suggests that our analyses regarding the ambient importance of SOA-INP are founded on the field measurements summarized in this figure and further supported by laboratory analyses summarized in Figures 3, S4, and others. We are

therefore appreciative of the reviewer's suggestions to clarify and strengthen the analysis of these results. To clarify the statistical analysis of Figure 2, we have made the following revisions to the text:

- a. We have first detailed how the regressions and uncertainty are derived. The following explanation has been integrated into Section 2.2 ("SOA Composition and INP Concentration"):

"Trendlines indicate ordinary least squares linear regressions. These are calculated using the filter-derived SOA mass loadings and the average INP concentration during filter collection. Regression coefficients and p-values for these least squares regressions are found in Table 1. Also illustrated via shading around trendlines are one standard error of uncertainty in the slope and intercept of the linear model. (Page 5 Lines 14-18)."

- b. We also thank the reviewer for highlighting the potential impact that the outlying datapoint at high INP concentrations may have on the analysis. To ascertain how this outlier biases the regression, we have also calculated ordinary least squares linear regressions after excluding this outlying data point. We have included correlation coefficients and p-values of the correlation coefficients in Table 1. To clarify this in the manuscript, we have also added the following explanation:

"Average INP concentrations and isoprene-derived SOA mass loadings were notably higher on Oct. 7th than other days. Since outliers can bias regressions and skew interpretations of atmospheric measurements (Mikkonen et al. 2019), we also tabulated regression coefficients for linear models derived with all datapoints except the high outlier (Table 1, in parentheses). Although this results in somewhat lower regression coefficients, p-values (> 0.05) for R^2 between INP concentration and isoprene-derived SOA mass loading and fraction indicate statistically significant correlations (Table 1). (Page 5 Lines 25-31)"

We thank the reviewer for suggesting Mikkonen et al. 2019 as a reference, which we have included in the text above.

- c. As the reviewer suggests, we also clarify the statistical analyses between INP concentration, meteorology, and gas phase concentration. Our revisions on these points are expanded upon in our response to comment 8 below.

3. What is the second set of regression coefficient values in parenthesis in Table 1?

The R^2 and p-values in parenthesis in Table 1 indicate values derived after excluding the outlying datapoint. We thank the reviewer for drawing our attention to this ambiguity, which we hope is now clarified in the text:

"Since outliers can bias regressions and skew interpretations of atmospheric measurements (Mikkonen et al. 2019), we also tabulated regression coefficients for linear

models derived with all datapoints except the high outlier (Table 1, in parentheses). (Page 5 Line 26-28)."

4. Why is 4-NP mass fraction well correlated with INP, but 4-NP concentration not? My guess is that this shows that you cannot make too bold conclusions based on this data.

The reviewer highlights an interesting result from our analysis. We believe the apparent correlation of 4-NP mass fraction may be an artefact of including the outlying data point in the regression analysis, as discussed in our response to comment 2.a above. We have included text to highlight this in the revised manuscript:

"The correlation coefficient between 4-NP and INP concentration is only 0.49 after excluding the outlying datapoint (Table 1). (Page 5 Lines 39-41)."

Further, we indicate that the apparent correlation between 4-NP mass fraction and INP concentration may in fact be due to co-occurrence between 4-NP and other INP sources, such as isoprene-SOA:

"The observed correlation could also be due to co-occurrence between different SOA types. For instance, the mass fraction of terebic acid and 4-NP correlate more strongly ($R^2 \geq 0.90$) with the mass fractions of the isoprene SOA markers than with INP concentration. (Page 5 Lines 41-44)."

5. The INP number in Fig 1 range from 0-70 #/L, but in Fig 2 and Fig S5 from 0-7#/L. Is only a subset of data presented in Fig 2 and S5 or is the data averaged differently? How does it affect your conclusions that the higher INP values are not included in the regression analysis?

Our correlation analysis in Figure 2 and S5 use the average INP concentration during the time that the filter samples were collected. These collection periods encompassed an entire day's INP concentration measurements. For example, on Oct. 7th, ambient INP concentration ranged from 2.6 to 19.6, with an average value of 6.7. Filter samples were only collected between October 7th through October 15th, which precludes the higher INP average daily INP values from October 5th and 6th from being included in our correlation analysis in Figure 2.

To clarify these points in the manuscript, we have added the following text:

"Aerosol filter samples were taken concurrently during INP measurements between October 7th and October 15th. (Page 4 Lines 45-46)."

"Measurements of gas-phase chemistry on site show little correlation with INP concentration averaged over daily measurement periods. (Page 5 Lines 6-8)."

"The average INP concentration during aerosol filter sampling correlates ($R^2 \geq 0.88$) with mass loadings of... (Page 5 Lines 13-14)."

“Figure S5 Caption: Each data point represents daily averaged values of INP and ACSM aerosol mass concentrations.”

6. The presented data set is very short, only 10 days. From Fig S1, I also understood that INPs were measured only during a short period on each day. It would be good to somehow show the times of the INP measurements also in the main manuscript, from Fig 1 you don't really see it.

We thank the reviewer for this helpful suggestion. Since the reviewer seems to find the indication of sampling time in Figure S1 helpful, we have replicated it in Figure 1 in the main manuscript. Shaded regions now indicate the times of day that depositional INP measurements took place.

We have also clarified this in the revised caption for Figure 1: *“Shaded regions correspond to periods of INP measurements.”*

7. Is the size distribution in Fig 1b averaged over the period of INP measurements or the whole day?

The size distributions indicated in Figure 1b indicate the average and standard deviation of aerosol size distributions during INP measurements. We have clarified this in the text and figure caption:

Figure 1 Caption: *“Average ambient submicron aerosol size distributions during INP measurements each day.”*

“Average aerosol particle size distributions during INP sampling are shown in Figure 1b. (Page 4 Line 22).”

8. Figure S1b and S3b. There is rather large variation in INP concentrations during each measurement period, while the meteorological and gas conditions are quite stable (as the time period is rather short), which probably results in these horizontal lines in the scatter plots. It would make sense to plot (and calculate R^2 values using) the average values of each INP measurement period against the average meteorological and gas concentrations during that time, or possibly show both 10min and sampling period average values. Perhaps using log-scale for INP concentrations would make the 10min plot more readable?

We thank the reviewer for their suggestion to improve this analysis. We agree that plotting average INP measurements against average meteorological and gas concentrations would allow for more direct comparison to Figures 2 and S5.

- a. To accommodate the new analysis, we have substantially revised Figures S1 and S3. These plots now shown both 10-minute averages and daily averages for all meteorological and gas phase variables. Also shown are ordinary least squares linear regressions for the daily average values. The R^2 and p-values for both 10

minute and daily average sampling periods are given in Tables S1 and S2. We have applied the reviewer's suggestion and use a log scale for INP concentration on the x-axis.

- b. Further, we have expanded our discussion on how we have performed these statistical analyses. In section S1 – Station Meteorology – we have included the following text:

“The average values of these variables during INP sampling are plotted against INP concentration in Figure S1a during 10-minute and daily averaged sampling periods. We have also calculated ordinary least squares linear regressions on INP concentration and meteorology. Regression coefficients and p-values for both 10 minute and daily sampling periods are recorded in Table S1.”

In section S4 – Station Gas Phase Chemistry – we have included similar details:

“The 10 minute and daily average values of these variables during INP sampling are plotted against 10 minute and daily average INP concentration in Figure S4a. Table S2 contains the regression coefficients and their p-values of ordinary least squares linear regressions to the data in Figure S4a.”

- c. We have also updated the figure captions of Figures S1 and S4 to reflect the newly added datapoints and trendlines.

9. In text you mention, that the INP number is connected to small aerosol number and organic mass fraction, and that INP size with organic mass fraction. Could you show the regression coefficient (maybe even a figure) between these variables (INP number and size with total aerosol number, small aerosol number and organic fraction)?

We again thank the reviewer for their suggestions to improve the statistical analysis on our datasets. We believe these additions will strengthen our study and support our finding of an SOA-derived source of INPs. We have performed a series of ordinary least squares linear regressions:

- (1) [INP] vs. [$P_D < 150$ nm]
- (2) [INP] vs. [$P_D > 150$ nm]
- (3) INP Diameter vs. Non-Refractory Organic Mass Fraction

We have made several changes to the manuscript to incorporate these analyses. First, we have added a figure to the Supplementary Information section illustrating correlations 1 – 3 above (Figure S2). We have also included an additional table recording the correlation coefficients, and their p-values, to the SI (Table S2). Finally, we have included more quantitative support to our claims in the manuscript. The following text has been revised:

- a. *“Sampling periods with abundant Aitken and small accumulation mode particles (diameters < 150 nm) correlate ($R^2 = 0.88$; p -value 1.93×10^{-5}) with periods with high average INP concentrations (Figure S2; Table S1). (Page 4 Lines 22-25).”*
- b. *“Conversely, average INP concentration is uncorrelated ($R^2 = 0.06$; p -value 0.46) with ambient concentrations of larger mode particles (diameters > 150 nm). (Page 4 Lines 25-26).”*
- c. *“Average INP diameter was statistically anticorrelated ($R^2 = 0.68$; p -value 0.002) with ambient non-refractory organic mass fraction (Figure S2; Table S2.) (Page 4 Lines 39-41).”*

10. On page 7, you explain that the trajectories with high INP concentrations are from forested areas with high isoprene emission. From FigS6 it seems that all red colored trajectories pass northern Italy with a lot of anthropogenic pollution as well (although it is hard to tell as the figure is rather small). Therefore, the statement on rows 14-16 might not be justified, as also other terrestrial gas and particle sources from both biogenic and anthropogenic emissions could explain the higher INP concentrations.

We thank the reviewer for highlighting the potential for anthropogenic emissions to influence our sampling. We have revised this text to acknowledge the possibility that high INP activated fractions from Northern Italy may be caused by these anthropogenic emissions:

“However, we note that the elevated fractional INP activation in air masses passing over Northern Italy may also be due to emissions of mineral dust, while anthropogenic INPs have been inferred to contribute for cirrus level temperatures. (Page 8 Lines 8-10).”

We cite here two references (Hoose and Möhler, 2012; Ullrich et al. 2017) which demonstrate the ability of these terrestrial and anthropogenic aerosols to activate as depositional INPs.

11. Did the measured ice residuals exhibit features of other types of SOA than isoprene-derived SOA, or did you only look for isoprene-SOA? Is there a reason why isoprene derived SOA is better ice nuclei than other SOA? You might want to comment on that in the article.

Our analysis of the composition of ice residuals relied on the unique marker of IEPOX SOA in PALMS mass spectra (Froyd et al. 2010). This means that we were unable to detect the presence of SOA material from other sources, such as α -pinene or anthropogenic volatile organic compound emissions, as these do not have identified unique signatures in PALMS. We have clarified this in the text:

“IEPOX SOA has a unique signature in PALMS mass spectra that enables the unambiguous detection of IEPOX SOA material in ice residuals. The other SOA sources considered here, such as α -pinene and anthropogenically-sourced SOA material, have no identified unique

signature. We were therefore unable to detect the presence of these other SOA materials in ice residuals. (Page 10 Lines 29-33)."

We propose the ice nucleation abilities of isoprene-derived SOA may be due to the material's relatively high viscosity, which has been previously suggested to promote heterogeneous ice nucleation (E.g. Murray et al. 2010). We expand on this in the text (references omitted):

"The relatively high glass transition temperatures of 2-MT, 2-MT OS, and other IEPOX organosulfates dimers and trimers potentially slows water diffusion into the particle. This is one phenomenon that can enhance the ice nucleation activity of SOA material. The observed ice nucleation ability suggests that the physiochemical properties of isoprene-derived organosulfates may explain their observed ice nucleation abilities. For instance, the formation of IEPOX-OS is accompanied by the rapid depletion of inorganic sulfates. This reduces hygroscopicity and increases aerosol viscosity, both of which are factors that could slow the uptake and diffusion of water into isoprene-derived SOA and potentially enhance their depositional ice nucleation properties. (Page 7 Lines 7-15)."

References:

Froyd, K.D., Murphy, S.M., Murphy, D.M., De Gouw, J.A., Eddingsaas, N.C. and Wennberg, P.O., 2010. Contribution of isoprene-derived organosulfates to free tropospheric aerosol mass. *Proceedings of the National Academy of Sciences*, 107(50), pp.21360-21365.

Murray, B.J., Wilson, T.W., Dobbie, S., Cui, Z., Al-Jumur, S.M., Möhler, O., Schnaiter, M., Wagner, R., Benz, S., Niemand, M. and Saathoff, H., 2010. Heterogeneous nucleation of ice particles on glassy aerosols under cirrus conditions. *Nature Geoscience*, 3(4), pp.233-237.

Rinaldi, M., Santachiara, G., Nicosia, A., Piazza, M., Decesari, S., Gilardoni, S., Paglione, M., Cristofanelli, P., Marinoni, A., Bonasoni, P. and Belosi, F., 2017. Atmospheric Ice Nucleating Particle measurements at the high mountain observatory Mt. Cimone (2165 m asl, Italy). *Atmospheric environment*, 171, pp.173-180.

Zhao, B., Wang, Y., Gu, Y., Liou, K.N., Jiang, J.H., Fan, J., Liu, X., Huang, L. and Yung, Y.L., 2019. Ice nucleation by aerosols from anthropogenic pollution. *Nature geoscience*, 12(8), pp.602-607.

Reviewer 2

The manuscript by Wolf et al presents a combined field and laboratory experimental study of ice-nucleating particles (INP) at cirrus conditions. The authors suggest that small organic aerosol particles originating from the atmospheric oxidation of isoprene are responsible for the observed ice-nucleation activity at the Puy de Dome observatory in France, based on aerosol composition and several types of correlations with INP concentrations. They conclude that this may be important for cirrus cloud formation and their climatic effects.

While I agree that there are indeed indications for the fact that isoprene-derived SOA may be a source of INP, the causality and the presented experimental evidence does not fully convince me. I acknowledge the different individual small pieces of evidence that may point in the right direction, but a straightforward and continuous proof is missing in my opinion. I cannot pinpoint to a potential fundamental flaw: I actually fully believe that the presented data have been obtained and analysed according to the highest standards.

So maybe I am too critical, but the manuscript's conclusion appear too speculative to me. Even the authors themselves often use conditional voice, see e.g., the summarizing sentence in the introduction on page 3, lines 44/45: 'This study presents atmospheric measurements indicating ambient biogenic SOA may be an important source of INPs.' Similar types of statements occur on page 4, lines 25/26 and 36/37.

We thank the reviewer for carefully reading our manuscript and considering our data and results. We understand and respect the reviewer's trepidations concerning our conclusions. Rather than presenting simple and direct evidence supporting the importance of isoprene-derived INPs in the ambient atmosphere, we establish our conclusions on a compilation of ambient measurements, statistical analyses, and laboratory studies.

We discuss our changes in response to the reviewer's concerns below. In many instances, we feel that changes in response to reviewers 1 and 3 also address this reviewer's concerns. If the reviewer can provide additional details on what they find too speculative, we will be happy to revise further.

We hope to convince the reviewer that this combination provides multiple lines of evidence supporting our hypothesis of SOA-sourced INPs. Field measurements of INP concentrations are still relatively rare for the reason that they involve complex instrumentation and a high level of user engagement (see e.g. Reviewer 1's opening remarks). Only a handful of studies have successfully characterized the composition of ambient INPs using online and offline analytical techniques, as we have done here (See e.g. Cziczo et al. 2012; DeMott et al. 2003, Cornwell et al. 2019). We therefore urge the reviewer to see value in our combination of field and laboratory measurements. Each independently provides evidence for the activity of isoprene-derived SOA in the cirrus cloud regime. Together, they demonstrate (1) a statistically significant correlation between isoprene-derived SOA mass loading and INP concentration in the ambient atmosphere, (2) experiments confirming the ability for isoprene-derived SOA to nucleate ice depositionally, and (3) measurements unambiguously identifying isoprene-derived SOA material in relatively pure (i.e. free from other INP material, like mineral dust, black carbon, or biologicals) ice residuals.

We had made several changes to the text in response to these and other reviewer's comments that strengthen the rigor of our analyses and more quantitatively imply a connection between ambient isoprene-derived SOA and INP concentration. For example, our revised statistical analysis concerning the correlations in Figure 2 more convincingly demonstrate a statistically significant link (R^2 and p-values, Table 1) between isoprene-derived SOA and INP concentration.

Below, we expand on and highlight several specific changes to the manuscript that address the reviewer's concerns.

I am not convinced by the explanations and experimental support provided for the suggested climatic importance of isoprene-derived SOA in cirrus clouds. The statement that '... our findings indicate that isoprene-derived INPs play an important role in the radiative budget.' (page 7, lines 19-21) to me is an exaggeration and, thus, highly speculative. Given this fact, I cannot confidently support publication. While the experimental data are indeed useful, the overall picture and interpretation is too weak to allow rating this work as 'representing important advances of significance' and 'providing strong evidence for its conclusions'. For a journal with such a high reputation as Nature Communications I would assume a more convincing line of evidence. I am sorry that I cannot be more positive at this time.

We thank the reviewer for emphasizing their concern about our exaggerated claims. In response, we have modified the language of our claims in several places. Specifically, we no longer explicitly claim that isoprene-derived INPs yield important impacts on ambient cirrus cloud formation and the radiative budget. The text now reads:

- a. Abstract: *"Here, we demonstrate the potential for biogenic SOA to activate as depositional INPs in the upper troposphere by combining field measurements with laboratory experiments. (Page 2 Lines 6-7)."*
- b. Abstract: *"Finally, we show that potential ambient concentrations of isoprene-derived SOA are likely high enough to be competitive with other INP sources. (Page 2 Lines 16-18)."*
- c. Abstract: *"This demonstrates that isoprene and potentially other biogenically-derived SOA materials could influence cirrus formation and optical properties. (Page 2 Lines 18-19)."*
- d. Atmospheric Implications: *"These results specifically highlight the potential impact of isoprene-derived organosulfate SOA on cirrus cloud formation. (Page 8 Lines 12-13)."*
- e. Atmospheric Implications: *"Given the rarity of INPs in the ambient atmosphere and the net warming effect of cirrus clouds on global climate, our findings indicate that isoprene-derived INPs may be abundant enough to impact cirrus nucleation. (Page 8 Lines 13-15)."*

We have removed references to the potential, but uncharacterized, impact on climate and the global radiative budget, and tempered language concerning the potential impact on cirrus cloud nucleation. We believe these statements are now fair and accurately summarize the implications of our work.

Scientific Points:

1. The strongest data appear to be the correlation between INP concentration and isoprene SOA mass loading presented in Figure 2. However, the pinene-derived SOA correlates strongly with isoprene SOA (actually stronger than with INP), so I was wondering what all these correlations really mean. Is it possible that the correlation with isoprene just indicates that aged INP (of whatever type) are more active than fresh ones, and they just happen to accumulate SOA material during their lifetime, and then preferably isoprene-derived SOA material given that isoprene is one of the most abundant biogenic volatile compounds in the atmosphere?

The reviewer raises good questions which we believe can be addressed with our data. We have made several clarifications and additions to the manuscript to address these points:

- a. First, we have improved and augmented our statistical analyses to clarify which SOA meaningfully correlate with INP abundance, and which do not. The reviewer correctly points out that while INP concentration correlates with α -pinene SOA mass loading, this does not necessarily indicate that α -pinene SOA material activated as INPs since α -pinene SOA correlates with isoprene SOA mass loading. This suggests α -pinene and isoprene could have been merely co-emitted. We acknowledge this in the text:

“The observed correlation could also be due to co-occurrence between different SOA types. (Page 5 Lines 41-42).”

Further, our new statistical analyses demonstrate that the correlation between α -pinene and INP abundance is not significant ($R^2 = X$; p-value = 0.22) after the outlying datapoint is removed. We have clarified this in the text:

“However, the correlation coefficient and p-value suggest a weaker and statistically insignificant correlation after excluding the outlying datapoint from Oct. 7th. This demonstrates that α -pinene SOA may not have been an important INP source during our measurements. (Page 5 Lines 37-39).”

By performing regression analysis with and excluding the outlying datapoint, we believe our results more convincingly demonstrate meaningful correlations between some SOA sources (i.e. isoprene-derived SOA), and demonstrate no indication of INP activation by others (i.e. α -pinene & our tested anthropogenic SOA sources).

- b. Second, we thank the reviewer for raising their point about aged or SOA-accumulating particles. We interpret this comment to ask: “How can we know whether the SOA themselves are nucleating, as opposed to say other well-known

sources of INPs that have just been aged and co-occur with SOA material, or have accumulated SOA material onto them?"

We have made several clarifications and additions to the text and SI section to clarify this point. First, we highlight our findings that the average diameter of INPs were often much smaller than would be expected for primary aerosols:

"The average INP diameter on days with elevated organic mass fractions was smaller (75 ± 24 nm) than on days with relatively smaller organic mass fractions (230 ± 42 nm). (Page 4 Lines 37-38)."

A diameter of 75 nm is indicative of a secondary INP source. This small diameter precludes the possibility that the INPs were aged primary particles or SOA-coated primary particles such mineral dust, biologicals, and black carbon (all of which are known primary aerosol INPs.)

Further, we have added new discussion to the text and performed additional statistical analyses indicating that INP concentration correlated with small, organic molecules in the ambient atmosphere:

"Aitken and small accumulation mode particles (diameters < 150 nm) correlate ($R^2 = 0.88$; p -value 1.93×10^{-5}) with periods with high average INP concentrations (Figure S2; Table S1). Conversely, average INP concentration is uncorrelated ($R^2 = 0.06$; p -value 0.46) with ambient concentrations of larger mode particles (diameters > 150 nm). (Page 4 Lines 23-26)."

"Average INP diameter was statistically anticorrelated ($R^2 = 0.68$; p -value 0.002) with ambient non-refractory organic mass fraction (Figure S2; Table S2). This indicate INP diameter was smaller on days with abundant SOA. (Page 4 Lines 39-42)."

We would like to highlight for the reviewer that we have included a new figure (Figure S2), which summarizes the data discussed in the cited text.

We believe these modifications address the reviewer's points. While we cannot (and do not!) claim that aged primary aged aerosol did not constitute at least some fraction of the measured ambient INP population, our analyses and data suggest that many INPs were not aged primary particles but rather small secondary organic aerosols.

(2) Page 5, lines 24-31: Apparently, the data also point towards other types of INP being important. Together with the ice residual measurements showing only ~9% of isoprene-related INP, I was wondering how important are the isoprene-derived SOA overall?

We thank the reviewer for raising this point. We have revised Figure 5 to more clearly illustrate the relative importance of isoprene-derived INP compared to other INP sources.

Figure 5 now includes both our calculations of potential isoprene-derived SOA INP concentrations as well as previous measurements of depositional INP concentrations in marine air and terrestrially-influenced air. Measured ambient INP concentrations in the former ranged from 0.2 to 3.3 L⁻¹, whereas concentrations averaged nearly 40 L⁻¹ in the Saharan air layer. In the continental free troposphere, typical depositional INP concentrations ranged between 1 and 10 L⁻¹ in a study by DeMott et al. 2003 This indicates that isoprene-derived SOA may be a competitive source of INPs in the ambient atmosphere, particularly in mineral dust poor regions.

We have amended the text to refer to the updated Figure 5 and clarify where isoprene-derived SOA is likely to matter as a source of INPs:

“Figure 5b illustrates typical depositional INP concentrations in marine (0.2 to 3.3 L⁻¹) and the mineral-dust rich Saharan air layer (0.2 to ~40 L⁻¹). Typical depositional INP concentrations in the free continental troposphere range from 0.1 to 10 L⁻¹. This indicates isoprene SOA may be an important fraction of depositional INPs in the ambient atmosphere, particularly in environments with low abundances of other INPs, such as mineral dust aerosol. (Page 7 Lines 32-36).”

Finally, the reviewer points out that Figure 2 illustrates a persistent INP population even at low or zero SOA mass fraction. This value demonstrates the relative importance of non-SOA sources to the INP population. We have clarified this in the text:

“One feature of these relationships is the persistent presence of INPs even at low SOA concentrations. This observation is illustrated by an INP concentration of 0.5 to 1.5 L⁻¹ even when SOA generation was not observed (Figure 1b) and SOA mass loading was below the detection limit. This is consistent with sampling diverse sources of INPs. INPs aside from SOA, such as mineral dust or other primary aerosols, likely constituted the baseline INP concentration during these periods. INP concentrations were enhanced as biogenic SOA became more abundant. (Pages 5-6 Lines 46-4).”

(3) Page 5, lines 38/39: ‘At these conditions, aged 2-MT OS – a major component of isoprene-derived SOA – has a glassy transition temperature of approximately 3 °C (Ref.40).’ As far as I understand the glass transition temperature quoted from Ref 40 refers to the pure component 2-MT OS, i.e. at dry conditions. At the cirrus-type conditions studied here ($S_{ice}=1.3$ corresponding to about RH=84%) the glass transition of 2-MT OS may be significantly lower.

We thank the reviewer for suggesting this clarification. Indeed, the glass transition temperature for 2-MT OS will be lower at an elevated RH. Herein we estimate the viscosity of the 2-MT OS at RH=84% based on the method described by previous studies, cited below We have made several changes to the manuscript to expand on our glassy phase calculations.

First, we have clarified in the main text that we estimate the viscosity and glassy state transition at experimental conditions:

“We estimate aged 2-MT OS – a major component of isoprene-derived SOA – to have a glassy transition temperature of 237_{-16}^{+10} K and a viscosity of value of 10^{12+0}_{-2} Pa s at these conditions (SI Material and Methods). These results suggest that organic particles containing a significant amount of 2-MT OS remain in a glassy or semi-solid phase state at conditions requisite for depositional ice nucleation. (Page 6 Lines 11-15).”

We have added further details to the supplemental information section:

The glass transition temperature and viscosity of the 2-MT OS at RH=84% are estimated based on the method described by previous studies (Zhang et al. 2019; DeRieux et al. 2018; Riva et al. 2019). Since no hygroscopicity measurements have been performed on 2-MT OS to our knowledge, we use the hygroscopicity of other biogenic organosulfates – limonene-derived organosulfate (L-OS) – as a surrogate for 2-MT OS. L-OS has a hygroscopicity parameter (κ) value of 0.03 for 100 nm particles at ~85% RH.

The viscosity of IEPOX-derived OS is calculated based on a modified version of the Vogel-Tammann-Fulcher (VTF) equation (Eqs. S1 & S2) by Angell et al.^{4,5}

$$\eta(\text{RH}) = \eta_{\infty} e^{\frac{T_0 D}{T - T_0}} \quad (\text{S1})$$

where η_{∞} is viscosity at infinite temperature and assumed to be 10^{-5} Pa s, T_0 is the Vogel temperature, T is the ambient temperature, and D is the fragility parameter that controls how closely a material follows the Arrhenius law (Angell 1995). When T reaches T_g , η reaches 10^{12} Pa s, a value commonly associated with glass transition. Then Eq. (1) becomes

$$\frac{T_g}{T_0} = 1 + 0.0255D \quad (\text{S2})$$

We then apply the Gordon-Taylor mixing rule to calculate the T_g of the 2-MT OS and water mixture. The Gordon-Taylor constant between IEPOX-OS and water, k_{GT} , is assumed to be 2.5 based on previous studies (Riva et al. 2019). The glass transition temperatures of 2-MT OS at dry conditions and water are notated as $T_{g, \text{dry}}$ and $T_{g, \text{water}}$ (136 K). The mass fraction of the 2-MT OS in the whole mixture is $w_{\text{org}}(\text{RH})$ and is a function of relative humidity (RH).

$$T_{g, \text{mix}}(\text{RH}) = \frac{(1 - w_{\text{org}}(\text{RH}))T_{g, \text{water}} + \frac{1}{k_{GT}}w_{\text{org}}(\text{RH}) T_{g, \text{dry}}}{(1 - w_{\text{org}}(\text{RH})) + \frac{1}{k_{GT}}w_{\text{org}}(\text{RH})} \quad (\text{S3})$$

The mass fraction of 2-MT OS at any specific RH can be estimated based on effective hygroscopicity parameter (κ):

$$m_{\text{H}_2\text{O}} = \frac{\kappa \rho_w m_{2\text{-MT OS}}}{\rho_{2\text{-MT OS}} \left(\frac{1}{a_w} - 1 \right)} = \frac{\kappa \rho_w m_{2\text{-MT OS}}}{\rho_{2\text{-MT OS}} \left(\frac{100}{\text{RH}} - 1 \right)} \quad (\text{S4})$$

where $m_{\text{H}_2\text{O}}$ and $m_{2\text{-MT OS}}$ are the masses of water and 2-MT OS. ρ_w and $\rho_{2\text{-MT OS}}$ are the densities of water (1 g cm^{-3}) and 2-MT OS (choose a value of 1.2 g cm^{-3} based on previous study³). The water activity and the relative humidity are a_w and RH, respectively.

Based on Eqns. (S1)-(S4) and the parameters chosen for 2-MT OS ($T_g=276 \text{ K}$, $k_{GT}=2.5$, $\kappa=0.03$, $D=10$), the glass transition temperature of 2-MT OS is estimated to be 237 K ($36 \text{ }^\circ\text{C}$). This temperature is warmer than the conditions at which SPIN operated ($-46 \text{ }^\circ\text{C}$). By including the uncertainty of D and κ values ($D=10(20)$, $\kappa=0.02(0.05)$) as lower and upper bounds), the glass transition temperature of 2-MT OS at SPIN's operating conditions ($T = -46 \text{ }^\circ\text{C}$, $\text{RH}_{\text{ice}} = 130\%$) is estimated to be $237_{-16}^{+10} \text{ K}$ and the viscosity is estimated to be 10^{12-2} Pa s .

These results demonstrate that the glass transition temperature of 2-MT OS was likely warmer than our experimental conditions ($-46 \text{ }^\circ\text{C}$), thereby allowing the particles remain glassy or semi-solid. This estimation supports our conclusion that 2-MT OS could remain sufficiently viscous to promote heterogeneous ice nucleation at cirrus conditions.

(4) Page 5, line 45 and following: The measurements on ice residuals were actually done in an entirely different (urban) location in a different year than those of the ambient aerosol and INP measurements. Moreover, using data of only 111 collected particles of which 10 exhibited features of isoprene-derived SOA (page 6, lines 1/2) to conclude that this 9% of particles support 'the causality of our correlation between isoprene SOA and depositional INP concentrations measured at the Puy de Dome observatory' appears to me as a bit of a stretch.

We have modified the text in two ways to address this comment.

- a. First, we have made more explicit that the ice residual analysis was done in a different location and time than the measurements of ambient INP concentration at the Puy de Dome:

"In separate experiments and sampling location, we measured the composition of depositional ice residuals (IRs) to provide more direct evidence of isoprene-derived INPs. (Page 6 Lines 28-29)."

- b. Second, we have removed our claim that the ice residual analysis "supports the causality" of the correlation observed at Puy de Dome. Instead, we conservatively

state that the finding of IEPOX SOA in ice residuals provides evidence that such particles activated as INPs in our summertime urban experiment.

“We measured that 9% of depositional IRs contained isoprene-derived SOA material. Moreover, these IRs did not contain signature of refractory material of other known primary INPs, such as mineral dusts, black carbon, and biological particles. The relatively pure SOA spectra demonstrate that IEPOX-SOA material from summertime urban air activated as INPs. (Page 6 Lines 39-43).”

Minor and Technical Remarks:

(5) In general, the manuscript is well written, although I felt the line of thought was somewhat hard to follow: the different pieces of evidence were presented consecutively, but only during the second read, I understood how they were related. Moreover, some of the details in the main text were not required for the general understanding (e.g., lines 29-32 on page 4).

We thank the reviewer for suggesting areas where the text could be altered to improve the readability of our manuscript. We have removed the indicated details to the Materials and Methods Section:

“The particle enrichment factor is size dependent, ranging from 1 (no concentration enhancement) to 25 for particle diameters between 50 nm and 1 μ m, respectively²⁸. (Page 9 Lines 22-24).”

“INP size was not measured directly. However, the INP-specific enhancement factor is indicative of the average INP diameter (Figure S2; SI Materials and Methods). (Page 9 Lines 39-41).”

We believe several other modifications we have made to the text improves the readability and helps weave together our multiple lines of evidence.

(6) Page 3, line 4: Change ‘whereas liquid water and mixed phase clouds’ to ‘whereas liquid water clouds and mixed-phase clouds’

We have made the indicated change:

“Whereas liquid water clouds and mixed phase clouds impart a net cooling effect on climate... (Page 3 Line 4).

(7) Page 3, line 37: do the given percentages refer to aerosol number or mass? Please specify.

We have clarified that these percentages refer to INP number:

“Other characterizations of ambient ice residuals suggest particles with SOA material constitutes 14 to 24% of the number of heterogeneous cirrus INPs. (Page 3 Lines 39-40).”

We also apologize for omitting a citation here. We have now included reference to Cziczo et al. 2013 and Froyd et al. 2010.

(8) Page 9, line 39: It appears to me that this reference is not correct, as in that paper the synthesis of 2-MT OS is not described at all, and that of 2-MT only as an intermediate product in another synthesis, i.e. of one of various IEPOX species. Please provide correct reference.

We thank the reviewer for drawing our attention to this error. We have included reference to Surratt et al. 2009, which more accurately and completely describes the synthesis and products of isoprene-SOA formation.

References:

Angell, C. A., Formation of glasses from liquids and biopolymers. *Science* 1995, 267 (5206), 1924-1935.

Cziczo, D.J., Froyd, K.D., Hoose, C., Jensen, E.J., Diao, M., Zondlo, M.A., Smith, J.B., Twohy, C.H. and Murphy, D.M., 2013. Clarifying the dominant sources and mechanisms of cirrus cloud formation. *Science*, 340(6138), pp.1320-1324.

DeRieux, W. S. W.; Li, Y.; Lin, P.; Laskin, J.; Laskin, A.; Bertram, A. K.; Nizkorodov, S. A.; Shiraiwa, M., Predicting the glass transition temperature and viscosity of secondary organic material using molecular composition. *Atmos. Chem. Phys.* 2018, 18 (9), 6331-6351.

DeMott, P.J., Cziczo, D.J., Prenni, A.J., Murphy, D.M., Kreidenweis, S.M., Thomson, D.S., Borys, R. and Rogers, D.C., 2003. Measurements of the concentration and composition of nuclei for cirrus formation. *Proceedings of the National Academy of Sciences*, 100(25), pp.14655-14660.

Froyd, K.D., Murphy, D.M., Lawson, P., Baumgardner, D. and Herman, R.L., 2010. Aerosols that form subvisible cirrus at the tropical tropopause. *Atmospheric Chemistry & Physics*, 10(1).

Riva, M.; Chen, Y.; Zhang, Y.; Lei, Z.; Olson, N.; Boyer, H. C.; Narayan, S.; Yee, L. D.; Green, H.; Cui, T.; Zhang, Z.; Baumann, K. D.; Fort, M.; Edgerton, E. S.; Budisulistiorini, S.; Rose, C. A.; Ribeiro, I.; e Oliveira, R. L.; Santos, E.; Szopa, S.; Machado, C.; Zhao, Y.; Alves, E.; de Sa, S.; Hu, W.; Knipping, E.; Shaw, S.; Duvoisin Junior, S.; Souza, R. A. F. d.; Palm, B. B.; Jimenez, J. L.; Glasius, M.; Goldstein, A. H.; Pye, H. O. T.; Gold, A.; Turpin, B. J.;

Surratt, J.D., Chan, A.W., Eddingsaas, N.C., Chan, M., Loza, C.L., Kwan, A.J., Hersey, S.P., Flagan, R.C., Wennberg, P.O. and Seinfeld, J.H., 2010. Reactive intermediates revealed in

secondary organic aerosol formation from isoprene. *Proceedings of the National Academy of Sciences*, 107(15), pp.6640-6645.

Vizueté, W.; Martin, S. T.; Thornton, J.; Dutcher, C. S.; Ault, A. P.; Surratt, J. D., Increasing Isoprene Epoxydiol-to-Inorganic Sulfate Aerosol (IEPOX:SulfInorg) Ratio Results in Extensive Conversion of Inorganic Sulfate to Organosulfur Forms: Implications for Aerosol Physicochemical Properties. *Environ. Sci. Technol.* 2019, 53 (15), 8682-8694.

Zhang, Y.; Chen, Y.; Lei, Z.; Olson, N. E.; Riva, M.; Koss, A. R.; Zhang, Z.; Gold, A.; Jayne, J. T.; Worsnop, D. R.; Onasch, T. B.; Kroll, J. H.; Turpin, B. J.; Ault, A. P.; Surratt, J. D., Joint Impacts of Acidity and Viscosity on the Formation of Secondary Organic Aerosol from Isoprene Epoxydiols (IEPOX) in Phase Separated Particles. *ACS Earth and Space Chemistry* 2019, 3 (12), 2646-2658.

Reviewer 3

This is an interesting and exiting paper. Wolf et al. set out field and laboratory evidence that biogenic naturally occurring secondary organic aerosol nucleate ice in the deposition mode whereas anthropogenic organic aerosol does not nucleate ice. This is exciting because it implies that changes in aerosol composition due to climate change, land use change and other human activities may affect the concentration of ice nucleating particles in the upper troposphere which may then be important for climate. There is evidence in the literature that glassy materials of atmospheric relevance can nucleate ice, but this paper is novel in that it uses data from the real atmosphere. The correlations on their [own] would be interesting, but inconclusive, but together with the size, composition and supporting lab work a convincing case is made. I support the paper's publication in close the current form.

We are grateful to the reviewer for their careful reading of our manuscript and accurately summarizing the potential implications of our findings. Moreover, the reviewer highlights several other implications, such as the potential impact on land use change on isoprene-SOA INP emissions. We address changes to the manuscript regarding these interesting and exciting ideas below (response to comment 6).

I have a few comments which I think will lead to further improvements in the paper:

1. Consider alluding the cloud type this paper is relevant for in the title. When I saw the title I immediately thought about mixed phase clouds (reflecting my bias). But, the idea that biogenic material might impact cirrus clouds is important and distinct, and therefore should be clear in the title. Possibly insert 'cirrus' before 'Ice nucleating particles'.

We thank the reviewer for suggesting more clarity in our title. We have adopted the reviewer's recommendation. The title is now:

"A Biogenic Secondary Organic Aerosol Source of Cirrus Ice Nucleating Particles"

2. P 5 ln 33-42 (and elsewhere). Yes, being in a glassy state correlates with a material's ability to nucleate ice. But, it is not the defining physical variable which controls if an aerosol will nucleate ice heterogeneous or if it will swell with water and freeze homogeneously (in fact, the glassy state is arbitrarily defined depending on the physical technique being used to probe a material). The key variable is diffusion of water. The diffusion coefficient determines how rapidly water will diffuse into a droplet, if it is slow relative the change in conditions, then a glassy aerosol will remain largely solid and then has the potential to nucleate ice. Diffusion is related to viscosity in a complex manner and the standard Stokes-Einstein equation breaks down near the glass transition (e.g. see [Price et al., 2016]). So, it is much more accurate to talk about aqueous aerosol with slow water diffusion having the potential to nucleate ice heterogeneous than aerosol with high viscosity or being in a glassy state.

We again thank the reviewer for this scientific clarification and have made several additions or changes to the text in response:

- a. Introduction: *“A high particle viscosity decreases the diffusion rate of water into an SOA particle, allowing it to remain glassy and potentially promote heterogeneous ice nucleation. (Page 3 Lines 33-35).”*
- b. Section 2.3: *“The relatively high glass transition temperatures of 2-MT, 2-MT OS, and other IEPOX organosulfates dimers and trimers potentially slows water diffusion into the particle. This is one phenomenon that can enhance the ice nucleation activity of SOA material (Page 7 Lines 7-9).”*
- c. Section 2.3: *“This reduces hygroscopicity and increases aerosol viscosity, both of which are factors that could slow the uptake and diffusion of water into isoprene-derived SOA and potentially enhance their depositional ice nucleation properties. (Page 7 Lines 12-15).”*

3. There are diffusion and viscosity measurements of the water-soluble component of SOA which should be drawn upon for this study [Price et al., 2015; Renbaum-Wolff et al., 2013].

We thank the reviewer for their comment and providing additional references that are helpful to our analysis. We agree that the results from Price et al., 2015 and Renbaum-Wolff et al., 2013 are useful in determining the diffusion timescale of water based on the viscosity of water soluble alpha-pinene SOA. The Price et al. 2015 paper highlights that water diffusion is much quicker than what the Stokes-Einstein equation predicted. Based on measurements of water diffusion in the water soluble alpha-pinene SOA using Raman spectroscopy, Price et al. demonstrated that water diffusion is 3-4 orders of faster than the calculations shown by the Stokes-Einstein equation at ~0.8 water activity.

In the revised version of this manuscript, we conducted further analysis to estimate the glass transition temperature and viscosity of 2-MT OS at cirrus conditions ($T = -46$ °C, $RH_{ice} = 130\%$), as shown above in the response to Reviewer 2’s Scientific Comment #3. The viscosity of 2-MT OS was estimated to be 10^{12+0}_{-2} Pa s at our measurement conditions, which is equivalent to a calculated diffusion coefficient of 10^{-24+2}_{-0} $m^2 s^{-1}$ by using the Stokes-Einstein equation (Renbaum-Wolff et al. 2013; Price et al. 2015; Shiraiwa et al. 2011). Based on the findings by Price et al., the water diffusion rate is $\sim 10^{-20+2}_{-0}$ $m^2 s^{-1}$. The timescale for water to diffuse through a 100-nm particle is estimated to $2 \times 10^{8+0}_{-2}$ s based on the equation provided by Renbaum-Wolff et al., which is much longer than the average time for ice nucleation to happen (~1000 s) (Knopf et al. 2018). Hence the particles will likely remain in a solid or semi-solid phase state during the course of ice nucleation.

Based on the above analysis, we revised the manuscript as follows:

Main Text:

However, Price et al., also shows that water diffusion could happen much faster than what the Stokes-Einstein equation predicted using the water soluble SOA material

similar to what was used by Renbaum-Wolff et al. (Renbaum-Wolff et al. 2013; Price et al. 2015). Although further studies on the diffusion rate of water within the 2-MT OS and isoprene derived SOA are needed, the diffusion rate of water within the 2-MT OS is estimated to be $\sim 10^{-20+2}_{-0}$ m² s⁻¹ based on the results provided by Price et al. This yields an average mixing time of $2 \times 10^{8+0}_{-2}$ s (SI Materials and Methods). The mixing time of water within the submicron 2-MT OS particle is much longer than the timeframe for the ice nucleation process (Knopf et al. 2018). This result suggests that water vapor is unlikely to melt the 2-MT OS particle at cirrus relevant conditions, and thus supports our evidence that particles containing a significant amount of 2-MT OS or isoprene-derived SOA have the potential to promote heterogeneous ice nucleation. (Page 6 Lines 15-25).

Supporting Information:

The diffusion timescale of water within the 2-MT OS particle can be calculated as follows. The viscosity of 2-MT OS was estimated to be 10^{12+0}_{-2} Pa s at the cirrus conditions, which is equivalent to a calculated diffusion coefficient of 10^{-24+2}_{-0} m² s⁻¹ using the Stokes-Einstein equation. Based on the findings by Price et al., the water diffusion rate is $\sim 10^{-20+2}_{-0}$ m² s⁻¹. The timescale for water to diffuse through a 100-nm particle is estimated to $2 \times 10^{8+0}_{-2}$ s based on the equation provided by Renbaum-Wolff et al.

4. P5. Ln 25. It is not clear what flattening out means when viewing these plots. I think it is simply that the Number of ice crystals does not tend to zero as SOA tends to zero.

We have clarified this text. We were trying to convey precisely what the reviewer suggests: that at low SOA mass loadings, INP concentration does not tend toward zero but rather reaches a minimum of 0.5 to 1.5 L⁻¹. The text now reads:

“One feature of these relationships is the persistent presence of INPs even at low SOA concentrations. This observation is illustrated by an INP concentration of 0.5 to 1.5 L⁻¹ even when SOA generation was not observed (Figure 1b) and SOA mass loading was below the detection limit. (Pages 5-6 Lines 46-2).”

5. P7. Ln 8. Ref [50] is a paper about mixed phase conditions and the comparison with the cirrus conditions made here should not be done. INP in the cirrus and mixed phase regimes will only sometimes be correlated.

We thank the reviewer for pointing out that DeMott et al. 2015 focused on mixed-phase cloud conditions. We have replaced the reference here with Ladino et al. 2016 and China et al. 2017, which measured low depositional INPs in a coastal and remote marine environments.

6. Could the authors suggest what might happen to alpha-pinene SOA in the future and what has happened to the balance between biogenic and anthropogenic SOA in the past? What are the implications for ice nucleation?

This is an interesting point, and we thank the reviewer for suggesting it. Climate change and land use/land cover change are forecasted to alter emission rates of a-pinene and isoprene, thereby impacting mass loadings of these SOA in the atmosphere. This may alter concentrations of INPs.

We have included a brief discussion of this in the “atmospheric implications” section (citations omitted):

“Further studies should quantify how evolving isoprene and other biogenic VOC emissions may impact INP concentration. Climate, land use, and land cover change have altered historical isoprene emission, resulting in a net isoprene decrease of approximately 25% since 1900. This emissions decline may have simultaneously led to a decrease in INP sourced from isoprene SOA. Future isoprene-SOA mass loadings may be similarly altered by evolving climate, CO₂ concentrations, land use patterns, and sulfate aerosol burdens. Additional modeling and experimental studies are needed to clarify the implications of these evolving SOA mass loadings for global INP concentrations. (Page 8 Lines 18-25).

7. Fig 1b. Is there a reason for plotting the size distribution in N rather than the more normal dN/dlogDp?

We apologize for the mislabeling. The correct units for Figure 1b’s y-axis are dN/dlogDp (cm⁻³). We have corrected the units in the new figure.

8. Consider bringing S4 and S6 into the main paper. I think the formatting would allow it and I think these plots are sufficiently important to the paper that they should be there.

We thank the reviewer for emphasizing the value of these figures. We have brought the original Figures S4 (laboratory fractional activation) and S6 (back-trajectories) into the main paper. They are now Figures 4 and 6 in the revised manuscript.

If the Editor deems this makes the paper too long, we will again move either or both figures to the supplement. If given a choice between the two, we would prioritize leaving the laboratory fractional activation figure in the main text over the back-trajectory analysis.

References:

China, S., Alpert, P.A., Zhang, B., Schum, S., Dzepina, K., Wright, K., Owen, R.C., Fialho, P., Mazzoleni, L.R., Mazzoleni, C. and Knopf, D.A., 2017. Ice cloud formation potential by free tropospheric particles from long-range transport over the Northern Atlantic Ocean. *Journal of Geophysical Research: Atmospheres*, 122(5), pp.3065-3079.

Knopf, D. A.; Alpert, P. A.; Wang, B., The Role of Organic Aerosol in Atmospheric Ice Nucleation: A Review. ACS Earth and Space Chemistry 2018, 2 (3), 168-202.

Ladino, L.A., Yakobi-Hancock, J.D., Kilhau, W.P., Mason, R.H., Si, M., Li, J., Miller, L.A., Schiller, C.L., Huffman, J.A., Aller, J.Y. and Knopf, D.A., 2016. Addressing the ice nucleating abilities of marine aerosol: A combination of deposition mode laboratory and field measurements. Atmospheric environment, 132, pp.1-10.

Renbaum-Wolff, L.; Grayson, J. W.; Bateman, A. P.; Kuwata, M.; Sellier, M.; Murray, B. J.; Shilling, J. E.; Martin, S. T.; Bertram, A. K., Viscosity of α -pinene secondary organic material and implications for particle growth and reactivity. Proc. Natl. Acad. Sci. USA 2013, 110 (20), 8014-8019.

Price, H. C.; Mattsson, J.; Zhang, Y.; Bertram, A.; Davies, J. F.; Grayson, J. W.; Martin, S. T.; O'Sullivan, D.; Reid, J. P.; Rickards, A. M. J.; Murray, B. J., Water diffusion in atmospherically relevant α -pinene secondary organic material. Chemical Science 2015, 6 (8), 4876-4883.

Shiraiwa, M.; Ammann, M.; Koop, T.; Pöschl, U., Gas uptake and chemical aging of semisolid organic aerosol particles. Proc. Natl. Acad. Sci. USA 2011, 108 (27), 11003-11008.

Reviewers' Comments:

Reviewer #1:

Remarks to the Author:

The reviewer comments have been addressed in a sufficient manner. I therefore recommend publication of the article subject to technical corrections.

Technical comment:

page 5, row 29. I think you mean $p < 0.05$ (smaller not larger)

I have now read the comments from Reviewer 2 and the answers and modifications by the authors.

I think the authors have provided sufficient answers to his/her scientific comments and clarified the text in places that he felt needed modification. However, there is no substantial new data presented, so it is hard to evaluate if Reviewer 2 would now be more convinced that the evidence is strong enough to support the conclusions. He does remark that he thinks the data is useful and obtained according to highest standards, so it is in the end a matter of opinion.

Reviewer #3:

Remarks to the Author:

The authors have addressed my previous comments. I have no further comments. I recommend that this paper is published.

Dear Editor,

We again thank you and our anonymous reviewers for their careful reading of our manuscript. Their further requests were minor, but are repeated below for clarity. Reviewer comments are in green text and our responses are in black.

Reviewer 1

The reviewer comments have been addressed in a sufficient manner. I therefore recommend publication of the article subject to technical corrections.

Technical comment:

page 5, row 29. I think you mean $p < 0.05$ (smaller not larger)

We thank the reviewer for catching this error! We have corrected this in the text (“ $p \leq 0.05$ ”).

I have now read the comments from Reviewer 2 and the answers and modifications by the authors.

I think the authors have provided sufficient answers to his/her scientific comments and clarified the text in places that he felt needed modification. However, there is no substantial new data presented, so it is hard to evaluate if Reviewer 2 would now be more convinced that the evidence is strong enough to support the conclusions. He does remark that he thinks the data is useful and obtained according to highest standards, so it is in the end a matter of opinion.

We thank the reviewer for their perspective.

Reviewer 3

The authors have addressed my previous comments. I have no further comments. I recommend that this paper is published.

We thank the reviewer for their helpful comments in the original review stage.